# SAM Audio: Segment Anything in Audio

Bowen Shi [* 1]  Andros Tjandra [* 1]  John Hoffman [* 1]  Helin Wang [* 1]  Yi-Chiao Wu [* 1]  Luya Gao [* 1]
Julius Richter [1]  Matthew Le [1]  Apoorv Vyas [1]  Sanyuan Chen [1]  Christoph Feichtenhofer [1]  Piotr Dollár [1]
Wei-Ning Hsu [1]  Ann Lee [1]

## Abstract

General audio source separation is a key capability for multimodal AI systems that can perceive and reason about sound. Despite substantial progress in recent years, existing separation models are either domain-specific, designed for fixed categories such as speech or music, or limited in controllability, supporting only a single prompting modality such as text. In this work, we present SAM AUDIO, a foundation model for general audio separation that unifies text, visual, and temporal span prompting within a single framework. Built on a diffusion transformer architecture, SAM AUDIO is trained with flow matching on large-scale audio data spanning speech, music, and general sounds, and can flexibly separate target sources described by language, visual masks, or temporal spans. The model achieves state-of-the-art performance across a diverse suite of benchmarks, including general sound, speech, music, and musical instrument separation in both in-the-wild and professionally produced audios, substantially outperforming prior general-purpose and specialized systems. Furthermore, we introduce a new real-world separation benchmark with human-labeled multimodal prompts and a reference-free evaluation model that correlates strongly with human judgment.

## 1. Introduction

Audio source separation aims to decompose a complex sound mixture into individual source tracks that correspond to distinct sound events. It serves as a critical tool for media production, accessibility, and multimodal AI research. Traditional *promptless* systems target fixed categories like speech or music (Mitsufuji et al., 2022; Zhao et al., 2024), but they struggle with open-domain mixtures where the boundaries between sound classes are ambiguous and highly context dependent. To address these limitations, *prompted* separation has emerged, using text (Liu et al., 2024b; Wang et al., 2025a) or visual cues (Zhao et al., 2018; Dong et al., 2023) to specify targets without relying on fixed taxonomies.

Despite this progress, significant challenges remain. Text-prompted models often underperform in specialized domains like music compared to dedicated tools such as Demucs (Rouard et al., 2023; Défossez, 2021), while visual-prompted methods are typically restricted to synthetic datasets. Furthermore, the field lacks unified benchmarks and reliable metrics. Standard objective measures like Signal-to-Distortion Ratio (SDR) require clean references that are rarely available in real-world recordings, and reference-free metrics like CLAP similarity (Wu et al., 2023) often show poor correlation with human judgment.

In this work, we introduce SAM AUDIO, a foundation model for general audio separation that unifies text, visual, and temporal prompting within a single framework. Built on a multi-modal diffusion transformer architecture trained with flow matching, SAM AUDIO allows users *what* to separate using a text description, *where* to separate using a visual mask via positive/negative clicks, and *when* to separate using a temporal span prompt. During inference, the model simultaneously produces target stem along with residual stem capturing all remaining audio content.

Our contributions include: **(1)** We propose SAM AUDIO, the first foundation model supporting multimodal prompting (text, visual, span) — used either individually or in combination, for open-domain separation, achieving state-of-the-art for both in-the-wild and professional audios of general and specialized domains (e.g., speech, music). **(2)** We introduce span prompting, a novel temporal conditioning technique that provides precise frame-level control over the separation process. **(3)** We develop SAM AUDIO-BENCH, a unified benchmark covering major audio domains with human-labeled multimodal prompts and a reference-free evaluation model that aligns closely with human perception.

---

[*]Equal contribution  [1]Meta SuperIntelligence Labs. Correspondence to: Bowen Shi <bshi@meta.com>, Andros Tjandra <androstj@meta.com>.

*Proceedings of the $43^{rd}$ International Conference on Machine Learning*, Seoul, South Korea. PMLR 306, 2026. Copyright 2026 by the author(s).

## 2. Related Work

**Speaker separation.** Traditional separation isolates sources by predicting masks in a latent domain. Architectures have evolved from early spectral masking (Takahashi & Mitsufuji, 2017; Luo & Mesgarani, 2018) to time-domain approaches like Conv-TasNet (Luo & Mesgarani, 2019), and subsequently to dual-path and Transformer-based models (Luo et al., 2020; Subakan et al., 2021; Zhao et al., 2024) that better capture long-range dependencies. Recently, the field has shifted toward generative modeling, leveraging diffusion and flow matching to handle complex overlaps (Mariani et al., 2023; Scheibler et al., 2025; Dong et al., 2025; Chen et al., 2023; Scheibler et al., 2023). While post-processing models can refine these outputs (Sawata et al., 2022; Wang et al., 2024), and enrollment-based methods extract speakers using reference audio (Delcroix et al., 2020; Wang et al., 2019; Ge et al., 2020; Wang et al., 2025b), text-prompted speaker separation remains underexplored despite its potential for flexible control without reference signals.

**Musical instrument separation.** Instrument separation typically relies on specialized architectures such as Demucs (Défossez et al., 2019; Défossez, 2021; Rouard et al., 2023) or sub-band Transformers (Lu et al., 2024; Kim et al., 2021). However, these systems are generally restricted to fixed taxonomies (e.g., the four stems of MUSDB18 (Rafii et al., 2017)), which limits their utility in open-domain scenarios and necessitates more flexible, promptable frameworks.

**Target sound separation.** To bypass fixed-ontology limitations (Kong et al., 2023), recent work leverages open-vocabulary prompting. Discriminative models like AudioSep (Liu et al., 2022) and generative approaches such as FlowSep (Yuan et al., 2025) align audio with text embeddings, yet often struggle to disambiguate subtle events in complex domains (Rouard et al., 2023). While visual cues provide instance-level grounding, from pixel-level associations (Zhao et al., 2018) to multi-level attention (Li et al., 2024b), they remain largely domain-specific or reliant on synthetic benchmarks (Huang et al., 2025), limiting real-world robustness. Unlike prior work using reference audio conditioning (Wen et al., 2025), we introduce *span prompting* to enable precise temporal control, bridging the gap between open-domain flexibility and grounded precision.

**Benchmarks and Evaluation.** Audio separation research primarily utilizes synthetic mixtures to facilitate reference-based evaluation. Speech datasets like WSJ0-2mix (Hershey et al., 2016) and WHAM! (Wichern et al., 2019) are foundational but lack realism, while conversational benchmarks like LibriCSS (Chen et al., 2020b) and CHiME (Barker et al., 2018) lack multimodal prompt coverage. Similarly, music and general sound benchmarks including MUSDB18 (Rafii et al., 2017), Slakh2100 (Manilow et al., 2019), and

AudioSep (Liu et al., 2022) target universal coverage yet remain largely synthetic. While video datasets like AVSpeech (Ephrat et al., 2018) and AVSBench (Zhou et al., 2022) introduce multimodal grounding, existing benchmarks typically focus on narrow taxonomies and evaluate only a single prompt modality.

Standard evaluation relies on distortion-based metrics such as SI-SDR (Vincent et al., 2006; Le Roux et al., 2019), which quantify energy differences rather than perceptual quality and often correlate poorly with human Mean Opinion Scores (MOS) (Cartwright et al., 2018). Although perceptual metrics like POLQA (Beerends et al., 2013) and data-driven predictors like NISQA (Mittag et al., 2021) attempt to bridge this gap, they generalize poorly to separation artifacts (Delgado & Herre, 2024). This highlights a critical need for evaluation frameworks that move beyond simple distortion errors to faithfully reflect human perception across diverse sound classes and prompt modalities.

## 3. Model

SAM AUDIO is a generative separation model that extracts both target and residual stems from an audio mixture conditioned on text, visual, and temporal span as prompts (see Figure 1). At its core, SAM AUDIO employs a flow-matching model built on a Diffusion Transformer (DiT) (Peebles & Xie, 2023) and operates in a DAC-VAE (Polyak et al., 2024) latent space to generate target and residual audio jointly.

### 3.1. Model Architecture

#### 3.1.1. FLOW-MATCHING WITH DIFFUSION TRANSFORMER

Traditionally, audio separation is treated as discriminative masked prediction (Liu et al., 2024b; Dong et al., 2023). However, to address the intrinsic *one-to-many* nature of isolating sounds from mixtures, SAM AUDIO adopts a generative flow matching paradigm (Lipman et al., 2023). This allows the model to learn the target distribution conditioned on multimodal prompts, similar to recent generation works (Le et al., 2023; Vyas et al., 2023; Polyak et al., 2024).

Our model learns a continuous vector field transporting a Gaussian prior $x_0 \sim \mathcal{N}(0, I)$ to the data distribution. At each step $t$, we predict the velocity field $u(x_t, c, t; \theta)$, offering better efficiency than standard diffusion models (Mehta et al., 2024). We employ a Diffusion Transformer (DiT) architecture (Peebles & Xie, 2023) where time embeddings modulate transformer blocks via *scale-and-shift* operations. A shared MLP maps $t$ to modulation parameters, reducing model size without compromising performance.

Audio is encoded into a compact sequence (25 Hz, $C = 128$) using a DAC-VAE (Polyak et al., 2024). By replacing

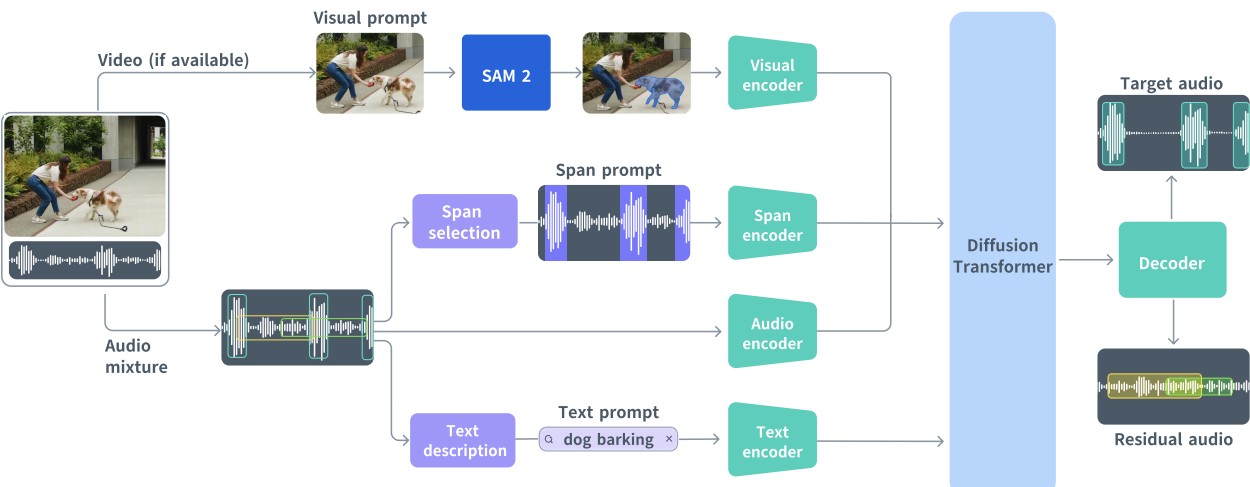

*Figure 1.* Overview of SAM AUDIO. Given an audio mixture, SAM AUDIO separates it into target and residual stems, conditioned on any combination of text descriptions (text prompts), visual masks (visual prompts), and temporal intervals (span prompts).

the standard residual vector quantizer with a VAE bottleneck (Kingma & Welling, 2013), we obtain smooth Gaussian latents ideally suited for continuous flow matching.

For SAM AUDIO, we jointly model target and residual sounds by concatenating their features $x = [x_{\text{tgt}}, x_{\text{res}}] \in \mathbb{R}^{T \times 2C}$. The model gradually denoises this joint representation, enabling simultaneous prediction of both components in a single pass.

### 3.1.2. PROMPT TYPES

SAM AUDIO supports three types of prompts: *text*, *span*, and *visual*, which can be used either individually or jointly to specify the target sound to extract.

**Text prompt.** The text prompt is a free-form natural language description of the target sound. For audio separation, we find that concise noun–verb phrase (NP/VP) descriptions (e.g., *"woman speaking"*) are more effective and natural for users than full sentences (e.g., *"a woman is delivering a speech"*). This format aligns better with common usage in audio editing.

**Visual prompt.** Video is a common source of multimodal audio data, and specifying a visual region offers an intuitive way to isolate a sound source. For example, to extract the sound of a barking dog in a video, the user can simply indicate the region of the dog in the visual frames (see Figure 1). We adopt SAM 2 (Ravi et al., 2024) to obtain the target visual mask. Given a video $V \in \mathbb{R}^{T \times H \times W \times 3}$, the user provides a binary mask $M \in \{0,1\}^{T \times H \times W \times 3}$ by interacting with SAM2 through clicks or bounding boxes. SAM AUDIO takes $(V, M)$ as input and, for simplicity, only processes the masked visual frames $V \odot M$. Note

visual prompt is specifically designed to extract sounds from onscreen objects due to its direct reliance on the masks. The separation is tightly coupled to the object's visible presence, naturally pausing if the target becomes occluded or moves off-screen.

**Span prompt.** Describing arbitrary audio events purely with text can be challenging, particularly for complex soundscapes such as movie soundtrack. To address this, we introduce *span prompting*, a form of temporal conditioning that specifies the time intervals during which the target sound occurs. The key intuition is that timestamps can disambiguate overlapping sound events. For example, in an audio mixture containing female speech from 0–6 s and dog barking from 1–2 s, the time interval alone suffices to isolate the barking sound. Formally, a span prompt is defined as a set of time intervals $S \in \mathbb{R}^{k \times 2}$, marking the target event start and end times.

### 3.1.3. AUDIO MIXTURE AND PROMPT ENCODING

To generate the target audio, SAM AUDIO conditions on both the input mixture and three prompts. The visual and span prompts are first encoded into frame-aligned feature sequences, combined with the latent representation from projected input audio features and noisy latent. In contrast, the text prompt is encoded into sequence embedding and integrated via cross-attention layers within the DiT backbone. The combination of frame-level and global semantic information help capture heterogeneous cues for separation.

**Audio encoder.** The input mixture is encoded with a separately trained DAC-VAE (Polyak et al., 2024), yielding a latent feature sequence $x_{\text{mix}} \in \mathbb{R}^{T \times C}$ at 25 Hz. We directly operate in the DAC-VAE space to minimize information loss

in audios, which ensures that the separated output remains faithful to the original audio.

**Text encoder.** We encode the text using a pre-trained `T5-base` encoder (Raffel et al., 2020), producing a token sequence of features $c_{\text{text}} \in \mathbb{R}^{N_{\text{txt}} \times d_{\text{txt}}}$, with $d_{\text{txt}} = 768$. These features are injected into the DiT backbone through cross-attention layers, enabling the model to focus on content that matches the description. As the prompts are short and unambiguous, we do not need to fine-tune the text encoder.

**Video encoder.** To provide visual grounding, we condition on frame-level features generated by Perception Encoder (PE) (Bolya et al., 2025). These visual features are resampled to match the audio latent features length $T$, mapped to the DiT dimension via a gated latent projection, and combined via elementwise-add operation with other modality latent representation.

**Span encoder.** Analogous to the phoneme sequence used in text-to-speech (TTS), we convert the span prompt, originally expressed as a set of time intervals $S \in \mathbb{R}^{k \times 2}$, into a frame-synchronous token sequence $S'_{1:T}$. Each token $s'_t \in \{\texttt{<sil>}, \texttt{+}\}$ denotes whether the target event is silent or active at frame $t$. The token sequence is embedded using a learnable embedding table and combined with elementwise-add operation with other modality latent representation. This provides an explicit temporal prior that guides the model to focus on the relevant regions for extraction.

Not all training examples have all three prompt modalities. In such cases, we provide dummy conditions: an empty string for text, an all-zero vector sequence for video, and a sequence of special null tokens (`<null>`) for entire span. To improve robustness, we also *randomly drop* each available prompt during training with probabilities $p_{\text{video}}, p_{\text{text}}, p_{\text{span}}$ and replace them with the corresponding dummy conditions. This encourages the model to robustly handle missing prompts at inference time.

### 3.2. Training Objective

**Flow Matching Objective.** At each flow step $t$, the model receives the noisy latent $x_t$ and conditioning set

$$c = \{x_{\text{mix}}, c_{\text{text}}, c_{\text{vid}}, c_{\text{span}}\},$$

where $c_{\text{text}}$ representing text features, $c_{\text{vid}}$ representing masked video features, and $c_{\text{span}}$ is the target audio span embedding. The model predicts the velocity field $u(x_t, c, t; \theta)$ used to update $x_t$ toward the clean target-residual representation

$$\mathcal{L}_{FM} = \|u(x_t, t, c; \theta) - (x_1 - (1 - \sigma_{min})x_0)\|, \quad (1)$$

where $x_1$ and $x_0 \sim \mathcal{N}(0, 1)$ are respectively target audio and random noise.

**Audio Representation Alignment.** To identify *what* and *when* to extract, we adapt the Representation Alignment (REPA) (Yu et al., 2025). We align intermediate representations with the embeddings of an external audio event detection (AED) model using auxiliary MLPs. This explicitly enforces event-level semantic and temporal consistency alongside the flow-matching objective.

For the alignment objective, we first extract the target audio embedding $a_{tgt} = \text{AED}(x_{tgt}) \in \mathbb{R}^{T \times F}$, where $F$ is the feature dimension of the AED module. In this work, we use pre-trained PANN-AED (Kong et al., 2020). Then, with $h_t \in \mathbb{R}^{T \times D}$ being the hidden DiT representation, where $D$ is the transformer's channel dimensionality, we project it to $\hat{a}_t = \phi(h_t) \in \mathbb{R}^{T \times F}$ where $\phi(\cdot)$ is a 3-layer MLP with 2048 hidden dimensions, GeLU activation and layer normalization. Then, we maximize the alignment between $a_{tgt}$ and $\hat{a}_t$ by minimizing

$$\mathcal{L}_{ali} = \mathbb{E}_t \left[ 1 - \frac{\hat{a}_t \cdot a_{tgt}}{\|\hat{a}_t\| \|a_{tgt}\|} \right]. \quad (2)$$

We combine both flow-matching loss and alignment loss into:

$$\mathcal{L} = \mathcal{L}_{FM} + \lambda * \mathcal{L}_{ali}$$

where $\lambda$ is a hyperparameter to control the influence of alignment loss.

### 3.3. Boosting text prompting with span prediction

Text prompts are accessible, yet span annotations offer precise frame-level control that enhances separation accuracy (see Section 6.4). To bypass the cost of manual labeling which can be time-consuming, we propose approximating spans during inference using $\text{PE}_{\text{A-Frame}}$ (Vyas et al., 2025), a model that predicts fine-grained event activity from text descriptions.

Given mixture $x_{\text{mix}}$ and prompt $c_{\text{text}}$, we threshold $\text{PE}_{\text{A-Frame}}$ outputs to generate approximate spans. Similar to Wang et al. (2022a), we then condition the separation model $\mathcal{M}$ on both the text and predicted spans. Formally, we perform text prompting through $\mathcal{M}(x_{\text{mix}}, c_{\text{text}}, \text{PE}_{\text{A-Frame}}(x_{\text{mix}}, c_{\text{text}}))$ instead of using it $\mathcal{M}(x_{\text{mix}}, c_{\text{text}})$.

### 3.4. Longform separation

While we mainly focus on one-shot audio separation, SAM AUDIO can be easily extended to long-form content using an iterative window-merging approach similar to Polyak et al. (2024), which avoids the boundary artifacts typical of independent chunkwise separation (see Appendix D.5 for details).

## 4. Data

SAM AUDIO is trained on tuples $(x_{\text{mix}}, x_{\text{tgt}}, x_{\text{res}}, c)$, where $x_{\text{mix}}$ is an audio mixture, $x_{\text{tgt}}$ the target stem, $x_{\text{res}}$ the residual stem, and $c \in \{\text{text}, \text{video}, \text{span}\}$ denotes one or more prompts. We construct audio tuples using three regimes (Table 1); when stems are unavailable, we either synthesize mixtures or bootstrap stems via pseudo-labeling. Overall, the raw data consists of a large-scale medium-quality audio–video corpus ($\sim 1M$) hours) and several high-quality datasets ($\sim 10K$ hours), covering speech, music, and general sounds (see Appendix A for more details).

### 4.1. Audios

*Table 1.* Overview of training data types used in SAM AUDIO.

| Category | Input Audio Mixture $x_{\text{mix}}$ | Target Audio $x_{\text{tgt}}$ | Residual Audio $x_{\text{res}}$ |
|---|---|---|---|
| Fully-real triplets | ✓ | ✓ | ✓ |
| Synthetic mixtures | ✗ | ✓ | ✓ |
| Pseudo-labeled stems | ✓ | ✗ | ✗ |

**Fully-real triplets** satisfy $x_{\text{mix}} = x_{\text{tgt}} + x_{\text{res}}$, providing the cleanest supervision. These are available for *music* ($\sim$500 hours of multi-track recordings) and *speech* ($\sim$21k hours of conversational corpora), where each channel contains a single speaker to provide ground-truth targets for natural overlaps.

**Synthetic audio mixtures** follow standard practices (Liu et al., 2022; Yuan et al., 2025; Ma et al., 2024) to augment limited stem data, combining diverse audio pools with complementary pseudo-labeling. *Noisy music* is synthesized by mixing music with general sounds at $\pm 15$ dB. *Noisy speech* is formed by mixing two-speaker speech with general noise, while *general sound mixtures* are created by combining in-the-wild clips and professional SFX to simulate multi-event environments.

**Pseudo-labeling** To mitigate the limitations of unrealistic random mixing, we bootstrap realistic training tuples from in-the-wild recordings using a two-stage separation and filtering pipeline. We first leverage PLM-Audio (Vyas et al., 2025) to generate event-based text prompts for a 1M-hour corpus, which then guide an intermediate SAM AUDIO checkpoint to produce candidate target and residual stems. To ensure label reliability, we apply a multi-stage filtering process using specialist models: CLAP (Wu et al., 2023) for text–audio alignment, Audiobox-Aesthetic PC score (Tjandra et al., 2025) for cleanliness, and ImageBind (Girdhar et al., 2023) for audio–visual correspondence. This engine effectively scales our training data across sound, music, and speech domains without manual stem annotations. Further implementation details, including specific filtering thresholds and data distributions, are provided in the Appendix A.

### 4.2. Prompt Generation

**Text prompts** are generated as concise noun/verb phrases (e.g., *dog barking*) rather than full sentences to more effectively specify target sound events. To scale production, the PLM-Audio captioning model generates initial descriptions which are merged with existing metadata using an LLM to produce cleaned, standardized event phrases. Final quality is ensured by filtering prompts through a CLAP-based similarity check, discarding samples where the audio–text correspondence falls below a 0.28 threshold (25th percentile).

**Visual prompts** enable instance-level separation by conditioning the model on a user-specified mask over video frames. For whole-video grounding, ImageBind-based filtering is applied to a 1M-hour corpus to remove non-diegetic content, such as off-screen narration or post-produced music, ensuring the audio is visually grounded. For pseudo-labeled data, visual masks are generated via SAM3 (Carion et al., 2025) using text captions as queries, with subsequent filtering to eliminate audio-visual mismatches or low-quality segmentations.

**Span prompts** utilize temporal intervals to specify target activity, making them particularly effective for discrete, "spiky" events like door slams or animal calls. These prompts are extracted using a voice activity detection (VAD) pipeline with a $-40$ dBFS threshold and a 250 ms minimum duration to identify active sound regions. This approach is primarily applied to high-quality isolated sound effects and multi-track music/speech, where temporal boundaries provide a strong signal for separating foreground events from continuous background ambience.

## 5. Evaluation

**SAM AUDIO-BENCH** Prior benchmarks often rely on synthetic mixtures or single-domain datasets, which lack the acoustic complexity and multi-modal alignment found in real-world environments. We introduce SAM AUDIO-BENCH, a benchmark sourced from 10 s in-the-wild clips (e.g., AudioSet (Gemmeke et al., 2017), AVSpeech (Ephrat et al., 2018), CondensedMovies (Bain et al., 2020)) that unifies speech, music, and general sound events under a consistent evaluation protocol. Each test item is densely annotated with: (i) language prompts normalized to concise NP/VP phrases, (ii) temporal boundaries (positive/negative spans), and (iii) per-frame visual masklets for on-screen sounding objects. This tripartite prompting allows for direct performance comparison across text, vision, and time modalities on identical audio mixtures. The benchmark consists of over 700 items, covering various separation tasks for music, speech and general sound effects in diverse acoustic environments. More details can be found in Appendix B.1.1.

**SAM Audio Judge** To address the limitations of simplistic,

reference-based evaluation metrics, we develop the SAM AUDIO-JUDGE (SAJ), a reference-free framework designed for fine-grained perceptual assessment of audio separation. We introduce a human annotation protocol evaluating different dimensions across separation model performance (Recall, Precision, Faithfulness, and Overall Quality). The SAJ model is a Transformer-based multimodal architecture that integrates pretrained audio and text encoders from PE-AV (Vyas et al., 2025) to predict these perceptual scores from the input mixture, model output, and text prompt. To improve robustness, the model is first pretrained on a large-scale proxy task of text-audio alignment detection (Wang et al., 2022b;a), and then finetuned on a diverse dataset of real and synthetic mixtures spanning speech, music, and sound effects, utilizing three independent human ratings per sample to ensure reliability.

**Metrics** We adopt human evaluation as the primary metric, utilizing a side-by-side Absolute Category Rating protocol to capture both absolute quality: overall score (OVR) and Net Win Rate (NWR). This hybrid framework empirically reduces uncertainty in score differences, narrowing confidence intervals by up to 20% compared to single-stimulus ratings. Objective performance for text and span prompts is assessed using SAJ and CLAP (Wu et al., 2023) scores to measure perceptual quality and semantic alignment. For temporal accuracy in span prompting, we introduce the SpanIoU metric, which calculates the intersection-over-union between predicted and reference spans. Finally, visual-prompted separation is evaluated via ImageBind (Girdhar et al., 2023) score. More details can be found in Appendix B.3.

## 6. SAM AUDIO Performance

### 6.1. Setup

We train three SAM AUDIO variants with 500M, 1B, and 3B parameters. Models follow the two-stage training recipe of Polyak et al. (2024), consisting of large-scale pre-training on synthetic mixtures followed by fine-tuning on curated high-quality and pseudo-labeled data. At inference time, we use a 16-step midpoint ODE solver with candidate reranking. We compare SAM AUDIO against a suite of SoTA open-source and proprietary models across speech, music and general sound effects. Below, we present the main results of SAM AUDIO. Further technical details regarding model configurations, baselines and ablation studies are deferred to Appendix C.

### 6.2. Text-prompted separation

Table 2 presents quantitative comparisons between SAM AUDIO and a range of public text-prompted separation models. The baselines fall into two broad categories: general

models that aim to handle a wide variety of separation tasks (such as AudioSep, FlowSep, SoloAudio, and CLAPSep), and specialized models optimized for specific domains like speech or music (e.g., MossFormer2, Demucs, Spleeter). In Table 2, we show the overall subjective score (i.e., OVR) aside from the objective metrics.[1] As our evaluation protocol follows a pairwise comparison setup, the final OVR score is obtained by averaging the overall preference scores across all pairwise comparisons involving SAM AUDIO.

Overall, SAM AUDIO consistently outperforms all prior models by a substantial margin across nearly all categories. In general sound event separation, SAM AUDIO achieves roughly a $\sim 36\%$ net win rate over one of the best public general sound separation model (SoloAudio (Wang et al., 2025a)). In specialized domains such as instrument or speaker separation, we observe that general-purpose models like FlowSep (Yuan et al., 2025) or AudioSep (Liu et al., 2022) fail to reach competitive quality. Their mean judge scores remain below 3, significantly trailing specialized systems such as Demucs (Rouard et al., 2023). Yet even in such domains, SAM AUDIO surpasses specialized models (e.g., NWR of SAM AUDIO vs. Demucs = 17.6%). While proprietary systems generally outperform the OSS counterparts, SAM AUDIO remains superior in most settings. On instrument separation for professional audios (MUSDB (Rafii et al., 2017)), SAM AUDIO achieves an overall subjective score of 4.45 compared to 4.28 from AudioShake (Audioshake, 2025) (a relative gain of $\sim 4\%$), and in speaker separation, it achieves 4.15 versus 3.51, corresponding to a net win rate improvement of $\sim 39\%$. Through unified training, SAM AUDIO generalize across domains and achieves SoTA performance.

### 6.3. Visual-prompted separation

We show the comparison between SAM AUDIO and existing visual-prompted separation models in Table 3. Compared to text-prompted separation, there are substantially fewer general-purpose visual separation systems available publicly.

Across all settings, SAM AUDIO achieves stronger improvements over prior work. On average, SAM AUDIO outperforms DAVIS by a large margin, achieving net win rates ranging from 5% to 48% depending on the separation task. Similar to the trends observed in text-prompted separation, we find that general visual models struggle on specialized domains such as *instrument* and *speaker* separation. In contrast, SAM AUDIO maintains and surpasses the best specialized baselines by approximately 25% on speaker separation and 5% on instrument separation.

---

[1]Net win rate against the baselines can be found in Appendix D.1.

*Table 2.* Comparison against text-prompted baselines. −: not applicable. OVR: overall subjective score.

| Model | OSS | Promptable | General SFX | | | Speech | | | Speaker | | | Music | | | Instr(wild) | | | Instr(pro) | | |
|---|---|---|---|---|---|---|---|---|---|---|---|---|---|---|---|---|---|---|---|---|
| | | | SAJ | CLAP | OVR | SAJ | CLAP | OVR | SAJ | CLAP | OVR | SAJ | CLAP | OVR | SAJ | CLAP | OVR | SAJ | CLAP | OVR |
| MossFormer2 (Zhao et al., 2024) | ✓ | ✗ | - | - | - | - | - | - | 2.43 | 0.14 | 2.54 | - | - | - | - | - | - | - | - | - |
| Tiger (Xu et al., 2024) | ✓ | ✗ | - | - | - | - | - | - | 2.47 | 0.15 | 2.50 | - | - | - | - | - | - | - | - | - |
| Fast-GeCo (Wang et al., 2024) | ✓ | ✗ | - | - | - | - | - | - | 2.66 | 0.16 | 2.71 | - | - | - | - | - | - | - | - | - |
| Demucs (Rouard et al., 2023) | ✓ | ✗ | - | - | - | - | - | - | - | - | - | - | - | - | - | - | - | 4.48 | 0.15 | 4.26 |
| Spleeter (Hennequin et al., 2020) | ✓ | ✗ | - | - | - | - | - | - | - | - | - | - | - | - | - | - | - | 4.26 | 0.11 | 3.90 |
| FlowSep (Yuan et al., 2025) | ✓ | ✓ | 2.36 | 0.21 | 2.65 | 2.18 | 0.20 | 2.14 | 1.85 | 0.09 | 2.13 | 2.73 | 0.18 | 2.90 | 2.37 | 0.10 | 2.69 | 2.13 | -0.01 | 2.02 |
| AudioSep (Liu et al., 2022) | ✓ | ✓ | 2.63 | 0.25 | 2.88 | 2.93 | 0.28 | 2.85 | 2.50 | 0.17 | 2.79 | 3.47 | 0.27 | 3.51 | 2.16 | 0.13 | 2.59 | 2.34 | 0.04 | 2.45 |
| CLAPSep (Ma et al., 2024) | ✓ | ✓ | 2.68 | 0.23 | 2.92 | 2.30 | 0.22 | 2.47 | 2.80 | 0.17 | 2.79 | 2.48 | 0.04 | 2.97 | 2.47 | 0.14 | 2.81 | 2.48 | 0.04 | 2.56 |
| SoloAudio (Wang et al., 2025a) | ✓ | ✓ | 3.29 | 0.25 | 2.97 | 3.45 | 0.30 | 3.32 | 2.26 | 0.19 | 2.45 | 2.68 | 0.21 | 2.47 | 2.92 | 0.13 | 2.71 | 2.65 | 0.01 | 2.30 |
| AudioShake (Audioshake, 2025) | ✗ | ✗ | - | - | - | 3.90 | 0.28 | 3.95 | 3.28 | 0.14 | 3.51 | 3.22 | 0.29 | 3.37 | 3.37 | 0.29 | 3.43 | 3.87 | 0.29 | 4.28 |
| MoisesAI (Moises.AI, 2025) | ✗ | ✗ | - | - | - | - | - | - | - | - | - | 3.79 | 0.27 | 3.90 | 3.03 | 0.29 | 3.12 | 3.78 | 0.28 | 4.22 |
| FADR (FADR, 2025) | ✗ | ✗ | - | - | - | - | - | - | - | - | - | - | - | - | 2.44 | 0.19 | 2.45 | 3.63 | 0.25 | 3.92 |
| LalalAI (Lalal.AI, 2025) | ✗ | ✗ | - | - | - | 3.77 | 0.33 | 3.92 | - | - | - | - | - | - | 3.07 | 0.25 | 3.03 | 3.83 | 0.27 | 4.18 |
| Auphonic (Auphonic, 2025) | ✗ | ✗ | - | - | - | 4.32 | 0.27 | 4.08 | - | - | - | - | - | - | - | - | - | - | - | - |
| ElevenLabs (ElevenLabs, 2025) | ✗ | ✗ | - | - | - | 3.79 | 0.25 | 3.72 | - | - | - | - | - | - | - | - | - | - | - | - |
| SAM AUDIO | ✓ | ✓ | 4.35 | 0.31 | 3.59 | 4.67 | 0.35 | 4.29 | 4.51 | 0.18 | 4.15 | 4.45 | 0.26 | 4.05 | 4.32 | 0.31 | 4.00 | 4.82 | 0.28 | 4.45 |

*Table 3.* Comparison against visual-prompted baselines. −: Not applicable. OVR: overall subjective score.

| Model | Generic | General SFX | | Speaker | | Instr (wild) | |
|---|---|---|---|---|---|---|---|
| | | IB | OVR | IB | OVR | IB | OVR |
| AV-MossFormer2 (Zhao et al., 2025) | ✗ | - | - | 0.20 | 2.62 | - | - |
| IIANet (Li et al., 2024b) | ✗ | - | - | 0.16 | 2.41 | - | - |
| ClipSep (Dong et al., 2023) | ✓ | 0.16 | 1.53 | 0.14 | 1.47 | 0.15 | 1.12 |
| DAVIS-Flow (Huang et al., 2025) | ✓ | 0.14 | 1.96 | 0.13 | 1.97 | 0.13 | 2.08 |
| DAVIS-Flow (Music) (Huang et al., 2025) | ✗ | - | - | - | - | 0.13 | 2.40 |
| SAM AUDIO | ✓ | 0.25 | 2.61 | 0.24 | 3.07 | 0.24 | 2.56 |

Additionally, we notice that visual prompting yields notably lower overall subjective scores compared to text prompting. The advantage of text prompt stems primarily from the availability of a much larger pool of high-quality text–based training data, whereas video-based supervision is generally noisier due to errors in visual masks and the frequent presence of off-screen sounds in training. Beyond scale, text also tends to provide more specific cues about the target source. For instance, a prompt such as *"man shuffling"* directly points to a unique sound event, whereas a visual mask of a person is inherently ambiguous since it can be associated with multiple sounds.

Nevertheless, visual prompting plays a complementary role in scenarios where text alone is insufficient. A common example arises in conversational scenes where multiple people of the same gender are speaking. A text prompt such as *"male speech"* cannot disambiguate between the two male speakers, but visual masks can localize the target speaker and enables separation effectively. In such cases, visual cues provide instance-level grounding that is difficult to achieve with text alone (see Figure 5 in Appendix).

### 6.4. Span-prompted separation

Table 4 compare models conditioned on *text*, *span*, and *text+span* inputs. To better ablate the effect of ground-truth span, no predicted span is used in text prompting. Using span prompts alone does not consistently improve performance, as the target and distractor sounds may co-occur throughout the same temporal regions. This effect is particularly evident for long-duration or ambient sounds such as speech and music, where span-only models exhibit large performance degradations (NWR of $-16\%$ to $-49.6\%$ relative to text-only baselines). In contrast, for short-duration and well-localized sounds, where temporal cues are more discriminative, span-only conditioning yields noticeable gains (NWR $+26\%$ over text-only in sound).

Despite these fluctuations, combining text and span inputs consistently improves performance across all domains, achieving NWR between $+12.9\%$ and $+39.0\%$. These results show that temporal localization from span prompts complements the semantic information in text, using both would enable more precise separation.

### 6.5. Span prediction boosts text-prompted separation

Leveraging predicted spans as additional input is our default choice for text prompting. Here, we compare using vs. not using span prediction across all separation tasks. The comparison is shown in Table 5. Note that the OVR scores for *text+predicted span* differ from those in Table 2, as the absolute ratings depend on the specific baseline used in each pairwise evaluation under our human evaluation protocol.

Incorporating predicted spans boosts performance in major-

*Table 4.* Comparison between span prompting and text prompting for SAM AUDIO. OVR: overall subjective score. For all the metrics below, higher is better.

| Prompt Modality | General | | | Speech | | | Speaker | | | Music | | | Instr(wild) | | |
|---|---|---|---|---|---|---|---|---|---|---|---|---|---|---|---|
| | SAJ | CLAP | OVR | SAJ | CLAP | OVR | SAJ | CLAP | OVR | SAJ | CLAP | OVR | SAJ | CLAP | OVR |
| text | 4.11 | 0.31 | 3.32 | 4.59 | 0.33 | 4.18 | 4.08 | 0.17 | 3.63 | 4.30 | 0.28 | 4.09 | **4.45** | 0.30 | 3.78 |
| span | 3.27 | 0.30 | 3.54 | 3.37 | 0.28 | 3.65 | 3.26 | 0.26 | 4.08 | 2.18 | -0.04 | 2.57 | 1.81 | 0.01 | 2.20 |
| text + span | **4.25** | **0.31** | **4.04** | **4.66** | **0.35** | **4.33** | **4.51** | **0.18** | **4.22** | **4.38** | **0.27** | **4.19** | 4.42 | **0.32** | **3.88** |

*Table 5.* Effect of using predicted temporal spans for text-prompted separation. OVR denotes overall subjective score; higher is better for all metrics.

| Task | w/ Pred Span | SAJ | CLAP | OVR |
|---|---|---|---|---|
| General SFX | ✗ | 4.11 | **0.31** | 3.36 |
| | ✓ | **4.35** | **0.31** | **3.89** |
| Speech | ✗ | 4.59 | 0.33 | 4.17 |
| | ✓ | **4.67** | **0.35** | **4.22** |
| Speaker | ✗ | 4.08 | 0.17 | 3.62 |
| | ✓ | **4.51** | **0.18** | **4.01** |
| Music | ✗ | 4.30 | **0.28** | **4.16** |
| | ✓ | **4.45** | 0.26 | 4.12 |
| Instr. (wild) | ✗ | **4.45** | 0.30 | 3.70 |
| | ✓ | 4.32 | **0.31** | **3.88** |
| Instr. (pro) | ✗ | **4.83** | **0.28** | **4.16** |
| | ✓ | 4.82 | **0.28** | 4.12 |

erate correlation, and signal-fidelity metrics like the SDR Estimator often fail to capture subjective quality, the proposed model's robustness across complex acoustic scenes highlights the effectiveness of its task-specific multimodal pretraining. More details on SAJ performance can be found in Appendix D.10.

*Table 6.* Overall Pearson (PCC) and Spearman (SRCC) correlation between automatic metrics and human ratings across different modalities.

| Model | Speech | Music | Sound |
|---|---|---|---|
| *Pearson Correlation* | | | |
| CLAP (Wu et al., 2023) | 0.490 | 0.487 | 0.367 |
| SDR Estimator (Dang et al., 2023) | 0.336 | 0.369 | 0.181 |
| Gemini-2.5-pro (Comanici et al., 2025) | 0.487 | 0.351 | 0.462 |
| **SAM Audio Judge** | **0.883** | **0.815** | **0.815** |
| *Spearman Correlation* | | | |
| CLAP (Wu et al., 2023) | 0.380 | 0.285 | 0.493 |
| SDR Estimator (Dang et al., 2023) | 0.338 | 0.390 | 0.173 |
| Gemini-2.5-pro (Comanici et al., 2025) | 0.495 | 0.338 | 0.390 |
| **SAM Audio Judge** | **0.817** | **0.714** | **0.781** |

ity of the domains, including general SFX, speech, speaker, and music, where temporal cues play a critical role in disambiguating target sounds. We noticed a minor degradation in professional instrument separation, likely because many MUSDB instrument stems span the entire segment, leaving limited room for temporal cues to provide additional benefit. There is only a small gap between using predicted spans and ground-truth spans (see Table 4), which shows the robustness of SAM AUDIO to temporal inaccuracies in span estimation. Importantly, span prediction improves separation quality without costly human annotations, allowing SAM AUDIO to separate precisely at scale.

### 6.6. SAM Audio Judge Performance

We compare SAJ model with representative baselines, including contrastive models (CLAP (Wu et al., 2023)), distortion-based measures (SDR Estimator (Dang et al., 2023)), and multimodal LLMs (Gemini-2.5-pro (Comanici et al., 2025)), across all evaluated domains. As shown in Table 6, SAJ achieves significantly higher alignment with human perceptual judgments, reaching Pearson correlation of 0.883 for speech and 0.815 for both music and sound. While baselines like Gemini-2.5-pro and CLAP show only mod-

## 7. Conclusion

We presented SAM AUDIO, a general-purpose separation model leveraging flow matching and multimodal prompting to achieve state-of-the-art performance. To overcome ground-truth scarcity, we introduced scalable pipelines for domain-aware synthetic mixing and pseudo-labeling from natural recordings. Beyond text, we introduced *visual* and *span prompting*, the latter of which significantly improves text-based results and enables iterative refinement. To support future research, we release SAM AUDIO-BENCH, a balanced multimodal benchmark, and SAM AUDIO-JUDGE, a reference-free metric that aligns more closely with human perception than existing alternatives. While visual grounding and complex sound effects remain areas for improvement, SAM AUDIO provides a robust framework for universal, grounded audio separation.

## Impact Statement

This paper explores advancements in unsupervised, array-agnostic, and generative blind speech separation, which lies in the intersection of several established fields, including array signal processing and machine learning. The algorithm we developed has many positive societal consequences. One notable application of our algorithm is assistive/augmented hearing, which improves communication and accessibility for individuals in noisy environments, especially those with hearing impairments. Also, our method has the potential to improve automatic speech transcription technology on ad-hoc microphone arrays. Although our method is generative, it does not synthesize harmful speeches that are not initially presented in the speech mixture. Nonetheless, the capability to isolate individual voices may pose privacy risks, which need careful regulations. We believe no other concerns require specific emphasis at this point.

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

# A. Data

Table 1 summarizes the three data construction regimes used in SAM AUDIO. Each regime differs in whether the mixture, target, and residual signals originate from real recordings or are obtained via synthesis or pseudo-labeling.

## A.1. Pseudo-labeling Data Engine

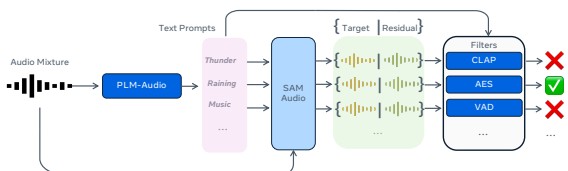

*Figure 2.* Illustration of our pseudo-labeling data synthesis pipeline. PLM-Audio generates text prompts from mixtures, which guide SAM Audio to produce target/residual stems. A filter stage retains only high-quality pseudo-labeled stems.

A limitation of the random mixing strategy described above is that it often produces mixtures that are unrealistic and poorly reflect real-world audio mixture. For example, mixing crowd cheering in a stadium with bird chirps recorded in a forest yields an unnatural combination that rarely occurs in the wild. Training on such mixtures can circumvent the model learning meaningful separation cues.

To address this issue, we synthesize more realistic training tuples by using an intermediate checkpoint of SAM AUDIO, trained with the identical recipe but without pseudo-labeled data. We generate target and residual stems from natural audio recordings using this early verison of SAM AUDIO, essentially bootstrapping new training data from unlabeled mixtures.

The data synthesis pipeline consists of two stages: *Separation* with SAM AUDIO and *filtering* various characteristics of the separated audio using specialist models.

**Separation.** In the separation stage, we run the intermediate SAM AUDIO checkpoint as a data engine to generate target and residual stems from large-scale unlabeled audio recordings, forming a pool of pseudo-labeled training candidates.

We focus primarily on text-based synthesis, given the scalability of text-prompted separation. We first apply the audio captioning model PLM-Audio (Vyas et al., 2025) to extract textual descriptions for the 1M-hour *in-the-wild* general sound data. The output of PLM-Audio is a list of audio event descriptions and we use each one as the text prompt for SAM AUDIO pseudo-labeling.

**Filtering.** In practice, this preliminary checkpoint exhibits significant variability in separation quality. To avoid degrading the final model, we apply strong filtering to remove low-quality candidates using a set of criteria covering var-

*Table 7.* Sources of training data for SAM AUDIO. "w/ stem" indicates whether ground-truth target/residual stems are provided.

| Data Source | Modality | w/ stem | Quality | Sound | Music | Speech | #Samples (M) | #Hours (K) |
|---|---|---|---|---|---|---|---|---|
| General Video | audio, video | ✗ | Medium | ✓ | ✓ | ✓ | $\sim 100$ | $\sim 1000$ |
| General Audio | audio | ✗ | Medium | ✓ | ✓ | ✓ | $\sim 1$ | $\sim 1$ |
| Speech Conversation | audio | ✓ | High | ✗ | ✗ | ✓ | $\sim 10$ | $\sim 10$ |
| HQ Music | audio | ✗ | High | ✗ | ✓ | ✗ | $\sim 10$ | $\sim 10$ |
| Multi-track Music | audio | ✓ | High | ✗ | ✓ | ✗ | $\sim 0.1$ | $\sim 1$ |
| HQ SFX | audio | ✗ | High | ✓ | ✓ | ✓ | $\sim 10$ | $\sim 10$ |
| HQ Video | audio, video | ✗ | High | ✓ | ✓ | ✓ | $\sim 0.1$ | $\sim 0.1$ |

ious quality aspects. We measure text–audio alignment using CLAP (Wu et al., 2023), assess audio cleanliness via the production-complexity (PC) axis of the Audiobox-aesthetic model (Tjandra et al., 2025), and detect overly silent outputs using a voice-activity detector (pydub). To enhance visual-prompt training, we further curate a subset with strong audio–visual correspondence. Given the same text prompt used for separation, we apply an in-house text-prompted video segmentation model to obtain visual masks, and calculates the audio-visual alignment score and mask coverage. For the former metric, we leverage Image-Bind (Girdhar et al., 2023), a large-scale audio–video–text contrastive model, to compute the cosine similarity between the embeddings of the audio track and masked visual frames. A pseudo-labeled sample is retained only if all of the following conditions in Table 8 hold.

*Table 8.* Filtering criteria for synthesized training data. Samples are kept only if all text–audio criteria are satisfied; visual-prompt samples require additional visual-related constraints.

| Criterion | Threshold |
|---|---|
| *Text–Audio Filtering (all must pass)* | |
| CLAP(text, target audio) | $> 0.35$ |
| CLAP(text, residual audio) | $< 0.0$ |
| Aesthetic PC score (target audio) | $< 2.5$ |
| Silence ratio (target audio) | $< 95\%$ |
| *Additional Visual-Audio Filtering* | |
| Mask coverage ratio (masked region) | $> 0.02$ |
| ImageBind(target audio, masked region) | $> 0.2$ |

Pseudo-labeling is applied to the general video portion of our training data (Table 7). We only run pseudo-labeling on mixtures that contain multiple sound events. After multi-stage filtering, the resulting pseudo-labeled set is substantially smaller than the high-quality audio used for synthetic mixtures. Table 9 summarizes the final pseudo-labeled datasets used for SAM AUDIO training.

## B. Evaluation Details

### B.1. SAM AUDIO-BENCH

#### B.1.1. SAM AUDIO-BENCH: UNIFYING MODALITIES, DOMAINS, AND REALISM

We observe three persistent gaps in existing separation benchmarks: (i) realism vs. references (in-the-wild acoustics vs. availability of stems), (ii) limited *prompt modality* coverage across text/visual/time, and (iii) *cross-domain* breadth under a unified protocol (speech, music, instruments, general sounds).

We introduce a **real, in-the-wild, multi-modal separation benchmark** addressing these gaps:

1. **Ecological validity**: All items are sourced from in-the-wild audio/video or production-quality video: **AudioSet** (Gemmeke et al., 2017), **VGGSound** (Chen et al., 2020a), **MUSIC** (Zhao et al., 2018), **MUSIC-AVQA** (Li et al., 2022), **AVSpeech** (Ephrat et al., 2018) and **CondensedMovies** (Bain et al., 2020).

2. **Multi-modal prompting**: Each 10 s test instance includes human annotated *visual* **SAM masklets** (Kirillov et al., 2023) (when sounding object is on-screen), *temporal* **positive/negative spans**, and *language* **text descriptions**.

3. **Taxonomic coverage**: Stem taxonomy is seeded from **AudioSet** ontologies (Gemmeke et al., 2017), with annotator-extendable classes. Dedicated tasks: **speech cleaning**, **speaker separation**, **music cleaning/removal**, **instrument stems** (37 classes), and **general sounds**.

**Annotation protocol.** Annotators enumerate sound events with text descriptions (that we subsequently normalize to concise noun/verb phrases), draw visual masklets for visible sounding sources – these masklets are available for every frame in the video, and every video is standardized to 24fps – and mark *temporal* presence/absence spans. Each (target sound, video) pair yields interchangeable text/vision/time

*Table 9.* Pseudo-labeled training data used in SAM AUDIO. Both audio-only and audio–video inputs are processed by an intermediate SAM AUDIO checkpoint to produce synthetic stems.

| Data Source | Modality | Synthetic Stems | Sound | Music | Speech | #Samples (M) | #Hours (K) |
|---|---|---|---|---|---|---|---|
| PL-Audio | audio | ✓ | ✓ | ✓ | ✓ | $\mathcal{O}(1)$ | $\mathcal{O}(1)$ |
| PL-Video | audio, video | ✓ | ✓ | ✓ | ✓ | $\mathcal{O}(0.1)$ | $\mathcal{O}(0.1)$ |

prompts, enabling controlled ablations across modality combinations.

### B.1.2. SUMMARY COMPARISON TO EXISTING EVALUATION SETS

Table 23 summarizes modality coverage, realism, domain scope, and reference availability across representative benchmarks versus our SAM AUDIO-BENCH.

As Table 23 shows, SAM AUDIO-BENCH uniquely combines (i) *real* in-the-wild audio/video, (ii) *multi-modal prompts* (text, visual masklets, temporal spans) on the *same* items, (iii) *cross-domain* coverage (speech, music, instruments, general sounds), and (iv) *reference-free human* evaluation.

Figure 3 shows summary statistics of the entire released SAM AUDIO-BENCH dataset, including how many test set items are available for each task, and which prompt modalities are supported for each; we also show a breakdown of which datasets SAM AUDIO-BENCH videos originate from.

### B.2. SAM Audio Judge Model

To develop SAM Audio judge, we conduct a systematic investigation into perceptually aligned evaluation for audio separation. Specifically, we aim to design an evaluation framework that (i) operates without requiring reference signals, making it suitable for real-world applications, (ii) enables fine-grained perceptual assessment of separated audio, and (iii) exhibits stronger correlation with human listening judgments.

### B.2.1. DATA COLLECTION

Existing evaluation guidelines for audio separation are typically simplistic, focusing only on coarse criteria such as the relevance between the separated audio and the prompt, or the overall audio quality of the output (Liu et al., 2022). Such high-level and loosely defined objectives make the evaluation ambiguous. Scores collected under these settings often conflate multiple perceptual aspects and may be biased toward certain criteria depending on the raters' individual interpretations, resulting in outcomes that are difficult to interpret consistently.

In addition, existing studies rarely examine the difficulty of audio separation tasks themselves. Most prior work has focused solely on evaluating model outputs, without systematically analyzing how intrinsic factors such as the number of overlapping sources, loudness imbalance, or acoustic similarity between sounds affect human perception of task difficulty. Understanding separation difficulty is crucial, as it provides insights into the limitations and robustness of current models, facilitates the curriculum design for training and evaluation, and enables difficulty-aware benchmarking and adaptive model selection in real-world applications.

To better characterize the performance of audio separation models and the intrinsic difficulty of audio separation tasks, we introduce a new human annotation guideline, **SAM Audio Judge (SAJ)**, which defines nine perceptual dimensions.

The SAJ performance dimensions evaluate how well a model separates the target sounds:

- **Recall**: Does the extracted audio contain all of the target sounds specified in the prompt?

- **Precision**: How effectively does the model remove non-target sounds from the extracted audio?

- **Faithfulness**: For target sounds present in the extracted audio, how similar do they sound to their counterparts in the original mixture?

- **Overall quality**: What is the overall perceptual quality of the model's output?

In addition, the SAJ difficulty dimensions assess the complexity of the separation task itself:

- **Counting**: How many non-target sounds are present in the source audio?

- **Overlapping**: To what extent do the target sounds overlap with non-target sounds?

- **Loudness**: How loud are the target sounds relative to the non-target sounds?

- **Confusion**: How easily can the non-target sounds be mistaken for the target sounds?

- **Overall difficulty**: Considering all the above factors, how difficult is it to extract the target sounds from the mixture?

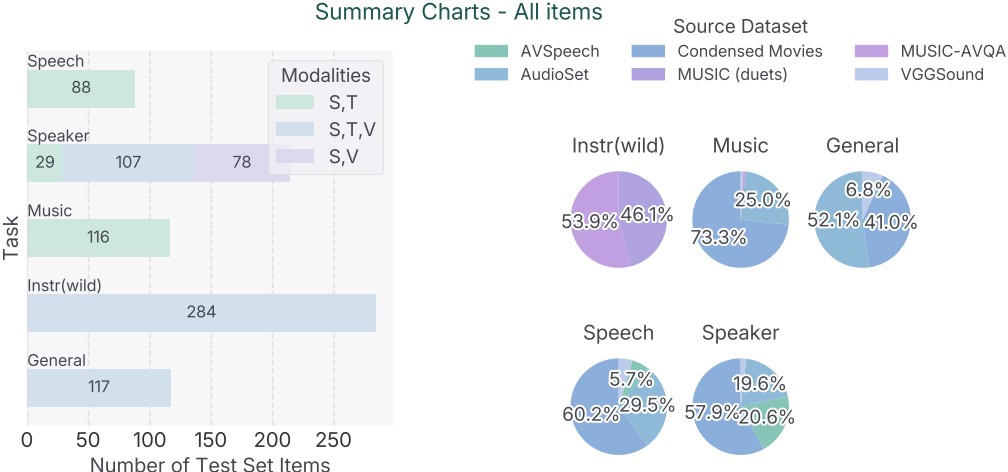

*Figure 3.* Summary of task, modality, and dataset coverage in SAM AUDIO-BENCH. The modality abbreviations are as follows: "T" indicates the item can be used with a text-only prompt (e.g. for speaker separation this implies that the text description can be unambiguously associated with a single speaker), "V" indicates that the target sound is on-screen and that we have a SAM masklet provided and "S" denotes that there are event boundaries for the target sound.

### B.2.2. DATA ANNOTATION

Based on the above definitions, we design an annotation task to collect SAJ data, where human raters evaluate the nine axes using a five-point Likert scale (1–5). We focus on **text-prompted** SAJ, the most commonly used scenario. We provide a comprehensive annotation guideline, which offers detailed explanations of each axis and specifies fine-grained aspects to consider during evaluation. To help raters calibrate their judgments, the guideline is supplemented with numerous audio examples and score references that illustrate what constitutes high or low scores along each dimension. We also design a rater qualification program to ensure the selection of high-quality annotators.

*Table 10.* Audio samples and duration in SAM Audio Judge DataSet

| Split | Modality | Duration | Samples |
|---|---|---|---|
| Training Set | Speech | 59.31 hrs | 13,149 |
| | Music | 133.64 hrs | 26,101 |
| | Sound | 117.52 hrs | 37,444 |
| Test Set | Speech | 6.38 hrs | 2,311 |
| | Music | 9.32 hrs | 3,367 |
| | Sound | 31.72 hrs | 11,476 |

**Paired data preparation.** We collect a comprehensive set of datasets spanning music, speech, and sound effects. To mitigate the mismatch between real-world and simulated data, we use both real mixtures and synthetic mixtures as input audio. Table 10 shows the detailed data statistics. For each data source, we adopt either the original sound annotations (e.g., dog barking, man speaking, etc.) or the sound type predictions generated by PLM-Audio as text

prompts. We adopt the same text prompt settings as the SAM Audio models, including speech extraction, speaker extraction, music extraction, instrument separation, and general sound event extraction. Following (Hai et al., 2024; Wang et al., 2025a), the text prompts are designed as single-sound descriptions, which may refer to either a specific sound source (e.g., dog barking, guitar) or a general sound category encompassing multiple sources (e.g., dog, music playing). For each modality, we gather outputs from various publicly available audio separation models as well as from our intermediate SAM Audio checkpoints, which are listed in Table 11. To establish a unified SAJ score that is calibrated across different audio modalities, we shuffle audio samples from all modalities during annotation. We also apply loudness normalization to eliminate potential confounding effects introduced by variations in audio volume. Finally, we collect three independent ratings per audio sample to reduce variance and improve reliability.

*Table 11.* Model outputs used in SAM Audio Judge DataSet

| Modality | Audio Separation Model |
|---|---|
| Speech | SAM Audio, MossFormer2, Tiger, FastGeCo |
| Music | SAM Audio, AudioSep, FlowSep, ClapSep, SoloAudio, Demucs, Spleeter |
| Sound | SAM Audio, AudioSep, FlowSep, ClapSep, SoloAudio |

### B.2.3. SAJ MODEL TRAINING

Our main SAJ model is designed to predict separation model performance, thus we only use the annotations along the performance dimension. As shown in Figure 4, the SAJ model takes three inputs: the input audio, the output audio, and a text prompt. We use a pretrained audio encoder to extract audio features and a text encoder to extract text features. Both encoders are adopted from PE-AV (Vyas

et al., 2025), which is trained through large-scale video–audio–text contrastive learning.

The text features are temporally aligned to match the length of the audio features and then fed into a Transformer to extract joint multimodal representations. Several linear layers are subsequently applied to predict the SAJ scores, including recall, precision, faithfulness, overall, and others.

In addition, we found that introducing a proxy task that predicts whether the output audio follows the text prompt (Wang et al., 2022b;a), significantly improves model performance. To this end, we pretrain the entire model on this text–audio alignment detection task using a large-scale simulated dataset that provides access to separated tracks within mixture audio. We alternate the output audio between the target sound and a random non-target sound from the same mixture to represent the presence or absence of the target sound. An additional linear layer is used to predict the presence or absence of the target sound described by the text prompt. After this pre-training stage, we finetune the SAJ model to predict the final SAJ scores.

### B.3. Subjective Evaluation Protocol

Given the limitations of objective metrics on in-the-wild content and the absence of reliable, general-purpose reference-free measures for separation quality, our benchmark adopts human evaluation as the primary method. We employ a **side-by-side Absolute Category Rating (ACR)** protocol with an always-on preference tie-breaker, a hybrid that yields both absolute quality signals and robust relative comparisons, while mitigating common evaluator biases.

Briefly, annotators are shown the source audio (and video, when applicable), the user prompt, and the extracted outputs from two models, and are asked to judge how well each output reflects the requested target sounds. The protocol begins by verifying whether the target sounds actually occur in the source audio, then assesses how much of the target content is present in the extracted audio, whether any portions are missing, and how similar the extracted target sounds are to their originals. Annotators also evaluate the presence and degree of non-target sounds, including whether they originate from the source audio or are artifacts introduced by the model. After answering these structured questions, raters assign a 1–5 Overall score reflecting fidelity to the prompt and acoustic faithfulness, followed—when model scores tie—by a forced-choice preference between the two outputs. The procedure applies consistently across text-, visual-, and span-prompted conditions.

Empirically, we find that this side-by-side ACR framework with an always-on preference tie-breaker offers clear advantages over alternative protocols. The tie-breaker improves inter-annotator agreement and yields sharper, more discriminative ACR deltas that align with expressed preferences. Side-by-side presentation also reduces uncertainty in score differences—narrowing confidence intervals by up to 20% compared to single-stimulus rating, which translates to roughly 30% cost savings for equivalent A/B sensitivity. Although handling time per item increases relative to pairwise-only preference evaluations, the protocol produces both absolute and relative judgments in a single pass. Finally, consistent with anchoring effects reported in prior work, absolute ACR scores remain context-dependent.

For each system pair, the subjective evaluation yields both an absolute score (OVR) and a net win rate (NWR) for system A vs. B across four dimensions: overall quality, coverage, correctness, and faithfulness. For our main result, we primarily report the overall OVR and NWR.

## C. Experimental Setup

### C.1. Model configurations

Table 12 lists the model configurations for three SAM AUDIO variants with parameter budgets (500M, 1B, and 3B).

*Table 12.* SAM AUDIO model configurations.

| Model | Total params | Layers | Attn dim | FFN dim |
|---|---|---|---|---|
| SAM AUDIO-SMALL | 500M | 12 | 1,536 | 6,144 |
| SAM AUDIO-BASE | 1B | 16 | 2,048 | 8,192 |
| SAM AUDIO-LARGE | 3B | 22 | 2,816 | 11,264 |

### C.2. Training details

**Pre-training.** We use an effective batch size of 1,024 sequences, each truncated or padded to 30 seconds ($=$ 750 audio tokens). Models are trained for 500K updates with a constant learning rate of $1 \times 10^{-4}$, preceded by a 5K-step linear warmup. AdamW is used throughout with weight decay 0.1 and `bf16` precision. During pre-training, the auxiliary alignment loss is enabled with weight 1. Additionally, we apply conditioning dropout: the audio mixture, video, and text prompt are each independently dropped with probability 0.3.

**Fine-tuning.** Since fine-tuning data exhibit higher length variability, we adopt *variable-length batching* with a per-batch token budget of {96K, 120K, 144K} tokens for the {500M, 1B, 3B} models, respectively. Fine-tuning runs for 300K steps with a 5K warmup to a peak learning rate of $1 \times 10^{-4}$, kept constant thereafter. An exponential-moving-average (EMA) checkpoint (decay 0.999) is maintained and used for inference. The auxiliary loss is disabled during fine-tuning (weight 0), as we do not observe gains when training on clean outputs. During fine-tuning, we also disable conditioning dropout.

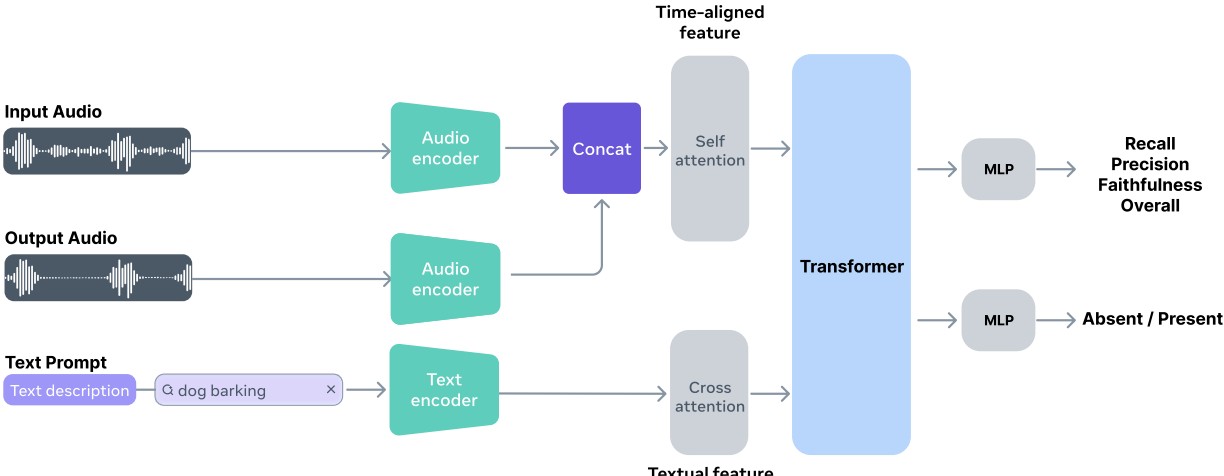

*Figure 4.* Diagram of SAM Audio Judge Model

**Inference.** We use a 16-step midpoint ODE solver without classifier-free guidance, as CFG yielded no improvement in our setting. We additionally apply candidate re-ranking with beam size 8: for text prompting, a linear combination of SAM Audio Judge and CLAP scores (weights 1 and 5); for span prompting, span IoU; and for visual prompting, the ImageBind similarity between audio and masked region. By default, we enable span prediction for text-only separation, as it brings overall gains (see Section 6.5). Specifically, we employ the PE-A frame model (Vyas et al., 2025) to estimate the temporal spans corresponding to the target sound given the text prompt. The predicted spans are then combined with the original text prompt and used as conditioning for SAM AUDIO. We set the frame probability threshold to 0.3, following the threshold setting in Vyas et al. (2025).

### C.3. Tasks

We benchmark SAM AUDIO across a diverse set of separation tasks that reflect common real-world use cases. Each task is to evaluate the model's ability to leverage text, video, and span prompts individually or jointly.

**Text-prompting tasks.** We evaluate SAM AUDIO on a broad range of text-prompted source separation tasks:

1. **General sound event extraction:** Extract any target sound event described by a text query (e.g., "dog barking" or "glass shattering"). This task tests the model's ability to handle extraction of general audio events.

2. **Speech extraction:** Separate all speech from a noisy audio mixture. The number of speakers is unconstrained, and the goal is to recover the complete speech track regardless of speaker count or background noise.

3. **Speaker extraction:** Isolate speech from a specific speaker given a text prompt describing speaker attributes such as gender or age (e.g., "female speaking"). This task evaluates fine-grained speaker conditioning. We primarily focus on gender-based separation.

4. **Music extraction:** Separate music from mixed audio, including cases where music is accompanied by speech or sound effects.

5. **Instrument separation in the wild:** Extract a single instrument stem (e.g., "piano," "drums") from full music audio. This benchmark includes both clean studio recordings and noisy in-the-wild music mixtures.

6. **Professional instrument separation:** Separate stems from professionally recorded music (e.g., studio tracks). Unlike the general instrument separation benchmark, this focuses on high-quality, multi-track recordings and uses MUSDB18 (Rafii et al., 2017) as the standard benchmark.

**Visual-prompting tasks.** Visual prompts consist of a pair of video masks and the raw video clip. These tasks evaluate the model's ability to perform instance-level source separation conditioned on visual information:

1. **General sound event separation:** Extract the sound associated with the highlighted region of interest (e.g., isolating the sound of car honking).

2. **Instrument separation:** In a music video, extract the audio of the instrument corresponding to the highlighted mask, regardless of whether the mixture is noisy.

3. **Speaker separation:** In a multi-speaker video, separate the speech of the highlighted speaker.

**Span-prompting tasks.** Span-prompting tasks mirror the text-prompting tasks. Instead of text descriptions, text model is conditioned on timespans that specify when the target source is active.

Within each modality, we filter out samples with ambiguous prompts that could map to multiple plausible sources (e.g., videos with multiple visually indistinguishable instruments or overlapping events that are not temporally resolvable). Each sample in our evaluation set are of $\sim 10s$, which is to facilitate subjective evaluation. As the original music in MUSDB span several minutes, we randomly extract 10s chunks of the original full-mix audio for professional instrument separation.

## C.4. Baselines

We compare SAM AUDIO against a broad set of baselines across all prompting modalities. For a fair comparison, we evaluate each model on tasks it is designed to handle, using its native input format where available.

### C.4.1. TEXT-PROMPTING BASELINES

Most existing research on text-guided audio separation focuses on general audio events, where broad categories such as *speech* and *music* are treated as single classes. We select four representative and recent open-domain baselines for comparison: AudioSep (Liu et al., 2022), FlowSep (Yuan et al., 2025), SoloAudio (Wang et al., 2025a), and CLAPSep (Ma et al., 2024). These models are designed to handle general audios and are trained on large-scale audio-text datasets. Although they are not specialized for individual domains, their pretraining data often include samples from speech and music, allowing a comparison to SAM AUDIO across general and specialized tasks.

Beyond these open-domain systems, we also include a range of specialized baselines targeting specific audio domains. Unlike general-purpose models, these systems do not support free-form text prompting; instead, they decompose an audio mixture into a fixed ontology of stems (e.g., vocals, drums, bass). For a fair comparison, we extract the separated stem that corresponds to the target event type and evaluate it against SAM AUDIO. The specialized baselines are outlined below.

**Instrument separation** We evaluate against Demucs (Rouard et al., 2023), Spleeter (Hennequin et al., 2020), AudioShake (Audioshake, 2025), MoisesAI (Moises.AI, 2025), LalalAI (Lalal.AI, 2025), and FADR (FADR, 2025). Demucs and Spleeter are publicly available models, while the others are proprietary systems

accessed via public APIs. All baseline models are limited to a small set of supported stems (see Table 13). For the professional instrument separation benchmark, we adopt MUSDB (Rafii et al., 2017), which only include vocals, drums and bass separation. In contrast, for instrument separation *in the wild*, we exclude Demucs and Spleeter since their supported stem vocabulary covers only a small fraction of required instruments. For the remaining baselines, we compare only on examples where the target instrument is within the supported list.

**Speech separation** Speech separation aims at removing background noise of a speech recording. We compare SAM AUDIO against LalaAI (Lalal.AI, 2025), ElevenLabs (ElevenLabs, 2025), Auphonic (Auphonic, 2025), and AudioShake (Audioshake, 2025). To ensure fair comparison, we disable post-processing or enhancement modules in the baselines, as our evaluation focuses on *separation fidelity*—the model's ability to isolate speech content—rather than perceptual enhancement or reverb suppression.

**Music separation** For music separation, we benchmark against MoisesAI (Moises.AI, 2025) and AudioShake (Audioshake, 2025), both of which can isolate or remove background music. Similar to speech separation, we disable any additional enhancement modules. We report results both for *music extraction* (isolating music) and *music removal* (removing music while preserving other sounds).

**Speaker separation** Speaker separation aims to isolate speech from specific individuals. Most specialized systems rely on fixed-stem decomposition and do not natively support prompting for arbitrary speakers. To enable comparison under the prompted setting, we select the separated stem with the highest CLAP similarity score (Wu et al., 2023). We evaluate public models such as Mossformer2 (Zhao et al., 2024), Tiger (Xu et al., 2024), and FastGeCo (Wang et al., 2024), as well as the proprietary AudioShake model (Audioshake, 2025).

### C.4.2. VISUAL-PROMPTING BASELINES

Compared to text-guided separation, visual-prompted audio separation is much less explored. No commercial models are available to our knowledge, thus we focus on public research models. We evaluate SAM AUDIO against two general-purpose visual separation models, DAVIS-Flow (Huang et al., 2025) and CLIPSep (Dong et al., 2023), across general sound, speaker, and instrument separation tasks. In addition, we include the instrument-specific checkpoint[2] of DAVIS-Flow (denoted as *DAVIS-Flow(Music)*) for the visual-prompted instrument separation benchmark.

Among visual separation tasks, speaker separation has been

---

[2] We use the DAVIS-Flow checkpoints trained on AVE and MUSIC respectively for general and instrument-specific separation.

*Table 13.* Supported instrument types for each baseline.

| Model | Supported Instruments |
|---|---|
| Demucs (Rouard et al., 2023) | vocals, drums, bass |
| Spleeter (Hennequin et al., 2020) | vocals, drums, bass, piano |
| AudioShake (Audioshake, 2025) | vocals, drums, guitar, bass, wind, piano |
| MoisesAI (Moises.AI, 2025) | vocals, guitar, bass, drums, piano, wind, strings |
| LalalAI (Lalal.AI, 2025) | vocals, drums, bass, guitar, piano, synthesizer, strings, wind |
| FADR (FADR, 2025) | vocals, drums, piano, guitars, strings, wind |
| SAM AUDIO | open-vocabulary |

studied more extensively (Wu et al., 2019; Pan et al., 2025; Li et al., 2024a;b). We therefore compare to two strong baselines, IIANet (Li et al., 2024b) and AV-Mossformer2 (Zhao et al., 2025), which achieve state-of-the-art results on a recent visual-prompted speech separation benchmark (Clear-Voice, 2025). These models rely on a preprocessing pipeline involving face detection and lip-motion extraction; we follow their official preprocessing setup and feed masked videos accordingly. In practice, about 20% of our evaluation video fail in AV-Mossformer2 preprocessing stage, and these samples are excluded from comparison with SAM AUDIO.

## D. Experimental Results

### D.1. Net Win Rates

Figure 6 and Figure 7 present the Net Win Rate (NWR) of SAM AUDIO against text-prompted and visual-prompted baselines, respectively. Figure 8 illustrates the NWR gains when utilizing temporal spans and joint text-span prompts compared to text-only baselines. Furthermore, the effectiveness of automated temporal alignment is shown in Figure 9, which compares the NWR of text combined with predicted spans against the text-only baseline.

### D.2. Visual-prompted separation samples

Figure 5 shows two samples of visual-prompted speaker separation.

### D.3. Sound Removal

SAM AUDIO outputs both a *target* and a *residual* audio. While earlier sections focus on evaluating target quality, we now evaluate the residual, corresponding to the *removal* task—removing the prompt-specified sound from the mixture. We use text-prompted music removal as a representative removal task.

Among existing baselines, only MoisesAI (Moises.AI, 2025) and AudioShake (Audioshake, 2025) support explicit sound removal. Therefore, we compare SAM AUDIO against these two systems. As shown in Table 14 and Fig-

ure 10, SAM AUDIO outperforms both systems. The trends mirror the music extraction results in Figure 6, suggesting the high correlation of extraction and removal modes. The high OVR score by SAM AUDIO further shows the model's ability to cleanly suppress target sources.

### D.4. Latency

Our default inference configuration uses 16 ODE steps with the midpoint solver. For text prompting, each separation on SAM AUDIO-Large takes approximately 7.3 s for a 10-second input on one A100 GPU, including 6.5 s for model forward computation, 0.1 s for span prompting, and 0.5 s for judge reranking.

We further study the trade-off between inference cost and output quality by varying the number of ODE steps. As shown in Figure 11, increasing ODE steps generally improves performance across tasks as expected. Nonetheless, the model achieves surprisingly competitive results even with as few as two ODE steps (e.g., speech separation). Qualitatively, we find a larger performance gap between 16 and 2 ODE steps for speaker or instrument separation. In contrast, for general sound effects that are often short and sparse, lower NFEs still yield acceptable perceptual quality. We hypothesize that the input audio mixture provides strong conditioning signal, enabling separation with fewer refinement steps compared to fully generative tasks such as TTS. Overall, fewer NFEs offer a favorable trade-off between speed and quality for many practical separation scenarios for SAM AUDIO.

### D.5. Long-form Audio Separation

To process long audio without memory issues or boundary artifacts, we adapt *multi-diffusion* (Bar-Tal et al., 2023; Polyak et al., 2024). We divide the mixture into overlapping windows, constructing conditions $c^{(j)} = \{x_{\text{mix}}^{(j)}, c_{\text{vid}}^{(j)}, c_{\text{span}}^{(j)}, c_{\text{text}}\}$ for each window $j$. At each flow-matching step $t_i$, we solve the ODE in parallel: $\tilde{x}_{t_{i+1}}^{(j)} = x_{t_i}^{(j)} + \Delta t_i\, u(x_{t_i}^{(j)}, t_i, c^{(j)})$. Local predictions are merged

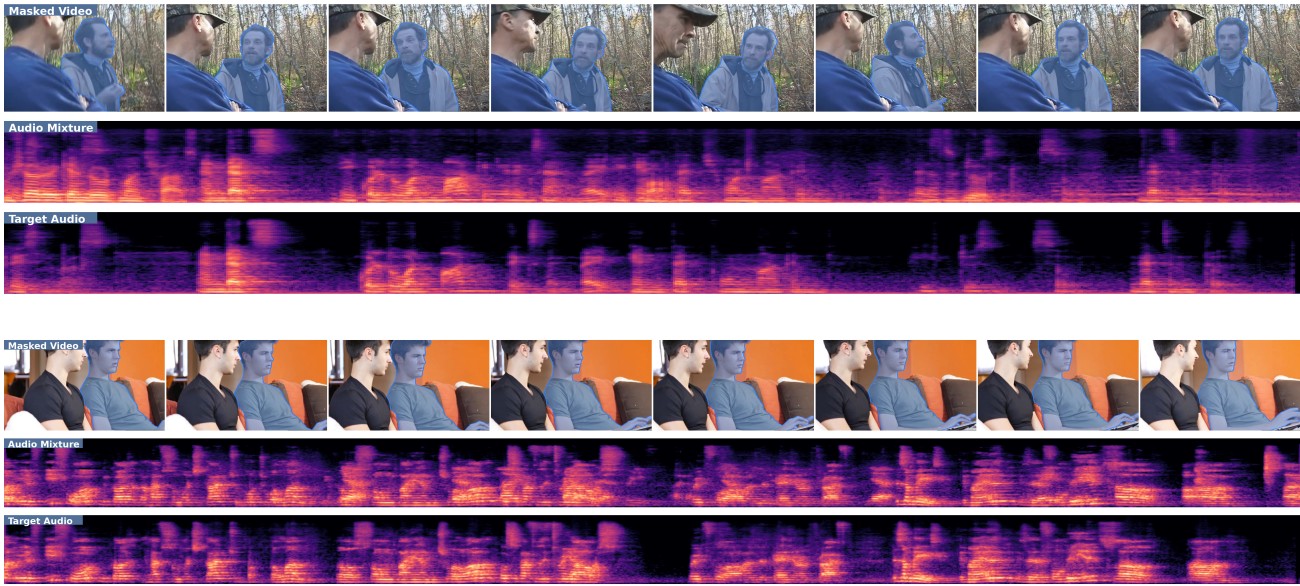

*Figure 5.* Visual-prompted samples where text descriptions are ambiguous. Each example shows (**top**) masked video, (**middle**) input mixture spectrogram, and (**bottom**) separated target spectrogram. The target speaker is highlighted in each video.

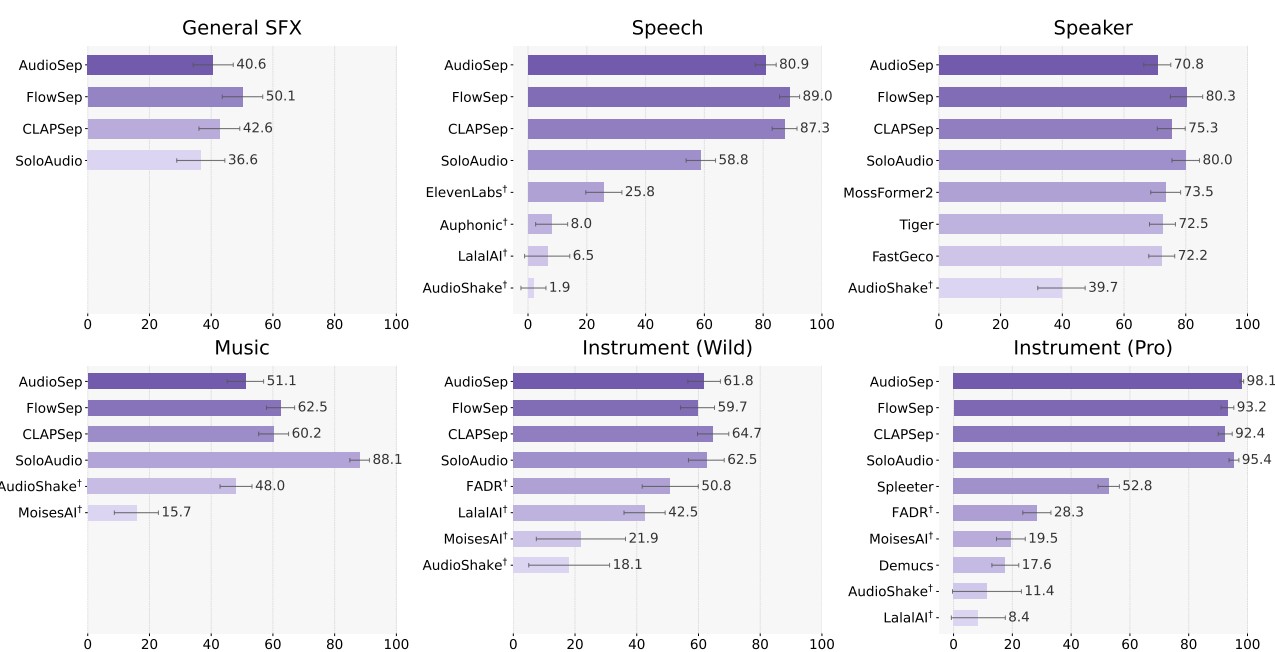

*Figure 6.* Net Win Rate (%) of SAM AUDIO against SoTA separation models in text-prompted tasks. †: proprietary models

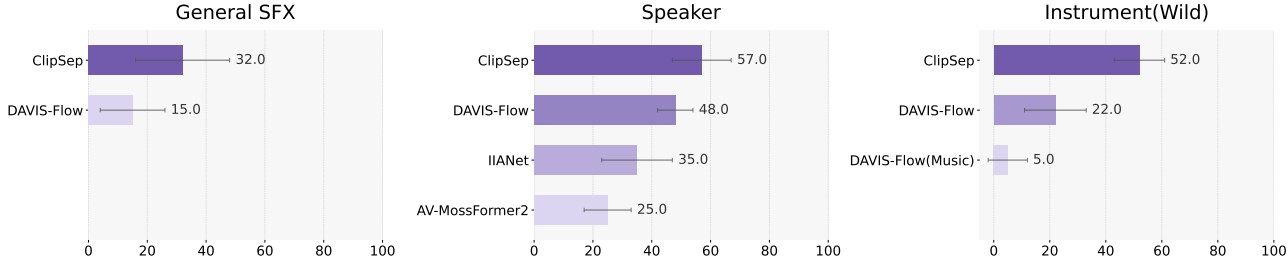

*Figure 7.* Net Win Rate (%) of SAM AUDIO against SoTA separation models in visual-prompted tasks

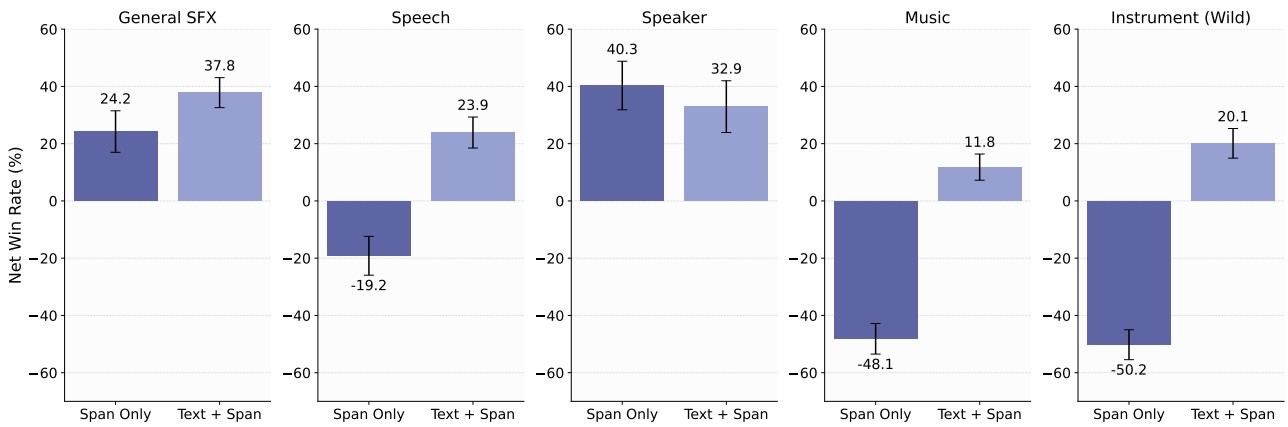

*Figure 8.* Net Win Rate (%) of SAM AUDIO with text & span / span as input against a text-only model

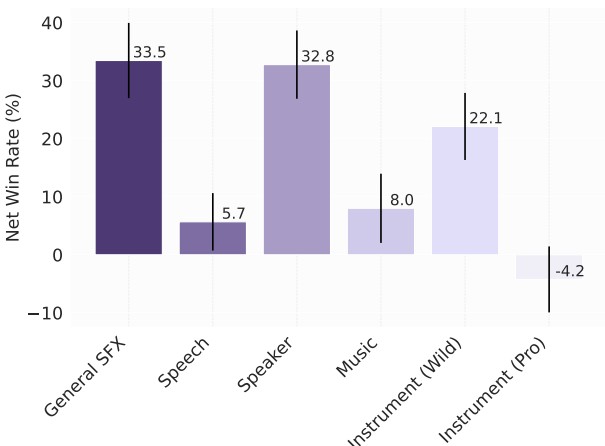

*Figure 9.* Net Win Rate (%) of SAM AUDIO using predicted span against not using predicted span for text prompting.

into a global latent using normalized soft masks:

$$x_{t_{i+1}} = \sum_j \text{pad}(m^{(j)} \odot \tilde{x}_{t_{i+1}}^{(j)}, j), \quad \text{s.t.} \sum_j \text{pad}(m^{(j)}, j) = 1,$$

where $m^{(j)}$ is a triangular window applied to segment $j$, and pad zero-pads it back to global length. This iterative merging ensures global coherence and eliminates discontinuities across segments.

Most samples in our evaluation set are around 10 seconds. We evaluate the performance of SAM AUDIO on long-form separation using the multi-diffusion approach. Specifically, we adopt a 20-second window with a 5-second context overlap. For comparison, we consider two baselines: (a) **chunkwise separation**, where the audio is divided into 20-second segments that are processed independently and stitched together, and (b) **one-shot separation**, where the entire audio is processed in a single forward pass.

We curate an internal test set of 50 1-minute audio samples,

*Table 15.* Comparison of methods for long-form separation.

| Method | SAJ | CLAP |
|---|---|---|
| One-shot | 3.48 | 0.26 |
| Chunk-wise | 3.57 | 0.24 |
| Multi-diffusion | **3.67** | **0.27** |

to evaluate long-horizon separation quality. As shown in Table 15, the one-shot model exhibits a noticeable degradation on long recordings, which aligns with expectations given that most training samples are shorter than 30 seconds. The chunk-wise baseline mitigates this issue but introduces audible discontinuities at segment boundaries. In contrast, the proposed multi-diffusion strategy maintains high perceptual quality across the full sequence and achieves the best judge scores.

### D.6. Effect of Model Scale

Tables 16 and 17, together with Figure 12, compare models of three sizes—500M, 1B, and 3B parameters. Across the scale comparison, we disable span prediction. Overall, the 3B model achieves the strongest performance across most tasks, though in certain cases such as general SFX separation it performs on par with or slightly below the smaller models.

Scaling model capacity provides the greatest benefit in specialized domains. For instrument separation, for example, SAM AUDIO-LARGE achieves substantial gains over the smaller variants: on instrument-in-the-wild separation, it outperforms SAM AUDIO-BASE by 23% NWR and SAM AUDIO-SMALL by 20%. These results suggest that larger models better capture the fine-grained acoustic cues required for high-fidelity separation in structured domains such as musical instruments, while smaller models remain competitive for broader sound categories.

| Model | OVR |
|---|---|
| AudioShake (Audioshake, 2025) | 3.75 |
| MoisesAI (Moises.AI, 2025) | 4.00 |
| SAM AUDIO | **4.05** |

*Table 14.* Comparison against baselines for music removal

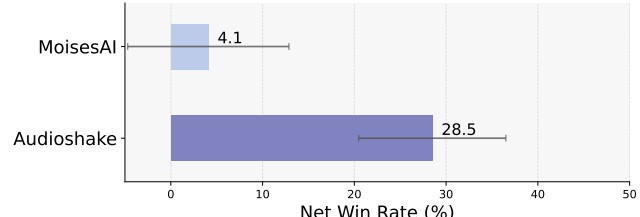

*Figure 10.* Net Win Rate (%) of SAM AUDIO over baselines in music removal

*Table 16.* Comparison of SAM AUDIO of different scales in text prompting. –: not applicable. OVR: overall subjective score.

| Model | General SFX | | | Speech | | | Speaker | | | Music | | | Instr(wild) | | | Instr(pro) | | |
|---|---|---|---|---|---|---|---|---|---|---|---|---|---|---|---|---|---|---|
| | SAJ | CLAP | OVR | SAJ | CLAP | OVR | SAJ | CLAP | OVR | SAJ | CLAP | OVR | SAJ | CLAP | OVR | SAJ | CLAP | OVR |
| SAM AUDIO-SMALL | **4.25** | 0.30 | **3.62** | 4.55 | **0.35** | 3.99 | 3.89 | **0.17** | 3.12 | 4.32 | **0.28** | 4.11 | 4.27 | 0.27 | 3.56 | 4.78 | **0.30** | 4.24 |
| SAM AUDIO-BASE | 4.23 | 0.28 | 3.28 | **4.61** | 0.33 | **4.25** | 3.94 | 0.15 | 3.57 | 4.26 | 0.27 | 3.87 | 4.33 | 0.29 | **3.66** | 4.78 | **0.30** | 4.27 |
| SAM AUDIO-LARGE | 4.11 | **0.31** | 3.50 | 4.59 | 0.33 | 4.03 | **4.08** | **0.17** | **3.60** | **4.30** | **0.28** | **4.22** | **4.45** | **0.30** | **3.66** | **4.83** | 0.28 | **4.49** |

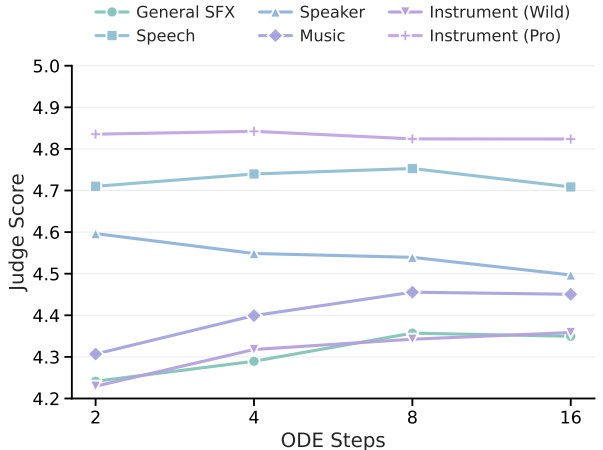

*Figure 11.* Effect of varying ODE steps under the midpoint solver. Fewer steps reduce computation at a modest cost in quality.

### D.7. Effect of Auxiliary Loss

Table 18 reports the effect of incorporating the representation alignment loss during SAM AUDIO pre-training. We compare two 3B pre-trained checkpoints: (1) a baseline model without the auxiliary loss, and (2) a model trained with the auxiliary loss using $\lambda = 1.0$. We evaluate both checkpoints on the general SFX test set under text- and visual-prompted separation, as the pre-trained model alone underperforms on specialized domains such as speech and music.

As shown in Table 18, adding the AED-based alignment objective yields consistent improvements in both settings, with a larger gain in the text-prompted case (over 20% relative improvement in text alignment) compared to visual

prompting ($\sim$5% relative improvement). This ablation is performed only during pre-training, as we do not apply the auxiliary loss in fine-tuning. We hypothesize that the benefit arises from the noisier audio target in pre-training corpus, where the alignment loss helps the model learn intermediate semantic representations beneficial for separation.

### D.8. Effect of fine-tuning

Table 19 compares the 3B model after pre-training only with the same model further fine-tuned on curated separation data. In addition to standard metrics, we also report the PC score from Audiobox-Aesthetics (Tjandra et al., 2025), which serves as a proxy for audio cleanness.

Overall, fine-tuning leads to consistent gains, though the magnitude varies across tasks. For text-prompted separation, the largest improvements appear in instrument extraction, speech extraction, and speaker separation. These tasks benefit from the availability of high-quality datasets containing professionally recorded stems, providing clean and reliable supervision. By contrast, improvements in general sound event separation and music separation are smaller. Large-scale pre-training already covers a wide distribution of sound events and mixtures, including many with background music, which leaves less room for fine-tuning to provide additional gains.

For visual-prompted separation, the visual alignment score (IB) remain relatively stable for general SFX, mirroring the trend seen in text-prompted separation. Fine-tuning data provide broader coverage for music and speech videos than for general sound events. Large-scale audio–visual pre-training already establishes strong correspondences between visual regions and audio, explaining the smaller incremental benefit for SFX.

*Table 17.* Comparison of SAM Audio of different scales in visual prompting. –: not applicable. OVR: overall subjective score.

| Model | General SFX | | Speaker | | Instr (wild) | |
|---|---|---|---|---|---|---|
| | IB | OVR | IB | OVR | IB | OVR |
| SAM Audio-Small | 0.24 | 2.62 | 0.23 | 2.79 | 0.21 | 2.25 |
| SAM Audio-Base | **0.25** | **2.63** | 0.24 | **3.25** | 0.22 | **2.76** |
| SAM Audio-Large | **0.25** | 2.61 | **0.24** | 2.95 | **0.24** | 2.58 |

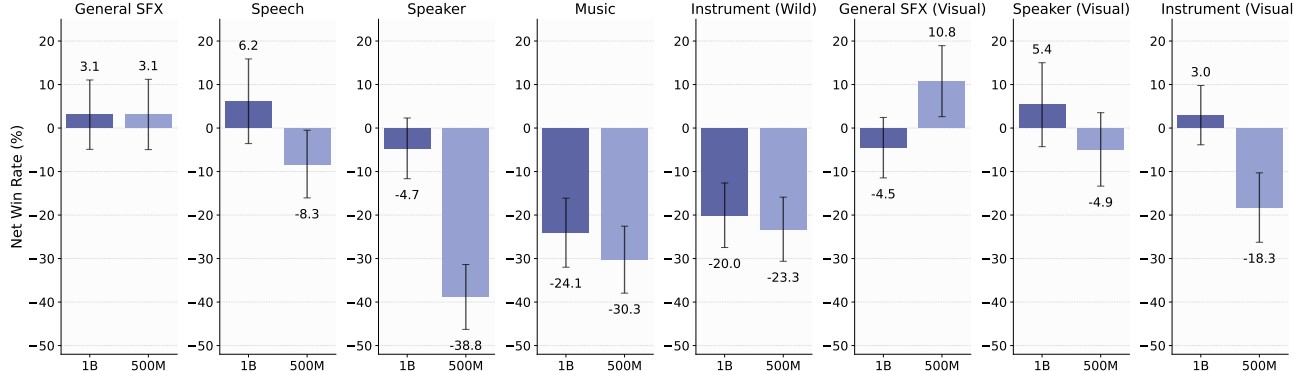

*Figure 12.* Net Win Rate (%) of SAM Audio-Base and SAM Audio-Small against SAM Audio-Large

*Table 18.* Effect of AED-based representation alignment loss in pre-training

| Auxiliary target | General SFX (Text) | | General SFX (Visual) |
|---|---|---|---|
| | SAJ | CLAP | IB |
| None | 2.48 | 0.24 | 0.17 |
| Target AED | **3.18** | **0.29** | **0.18** |

Finally, we observe a substantial improvement in audio cleanness across tasks, according to the PC metric. Fine-tuned models consistently produce cleaner separated audios with fewer artifacts, owing to the clean audio targets used during fine-tuning.

## D.9. Effect of using pseudo-labeled audio stem data

Table 20 shows a comparison of using or not using pseudo-labeled audio data in the fine-tuning stage. Details of the pseudo-labeled data are summarized in Table 9. The baseline model is trained using only fully real triplets and synthetic mixtures. As shown in Table 20, incorporating pseudo-labeled data yields consistent gains for both text and visual prompting. Notably, the largest improvements are observed in AES-PC for general sound, indicating that pseudo-labeled audio helps the model learn to produce cleaner separation stems.

## D.10. SAM Audio Judge Results

### D.10.1. SUBJECTIVE SCORE CORRELATION

We compare the proposed SAM Audio Judge model with several representative baseline systems covering diverse evaluation paradigms:

- CLAP (Wu et al., 2023): A large-scale contrastive audio–text model trained on millions of audio–caption pairs. We use its cosine similarity between the text prompt and output audio embeddings as a proxy for perceptual alignment. This represents the current standard for audio–language correspondence evaluation.

- SDR Estimator (Dang et al., 2023): A regression model trained to estimate SDR without access to ground-truth references. It reflects the performance of traditional distortion-based metrics that focus on signal fidelity rather than perceptual judgment. We trained the SDR estimator using the same architecture as the SAJ model, except that its training target was the SDR value. We used a balanced training dataset covering target audio levels from $-25$ dB to $25$ dB, totaling $496$ hours across speech, music, and sound effects. The SDR estimator achieved PPC scores of 0.923, 0.681, and 0.665 on speech, music, and sound effects, respectively.

- Gemini-2.5-pro (Comanici et al., 2025): A large multimodal LLM capable of reasoning over both text and audio inputs. We prompt it to rate the separated audio quality according to the same evaluation axes used in SAJ, representing a language-model-based perceptual

*Table 19.* Comparison of pre-trained vs fine-tuned results across audio separation tasks. –: not applicable.

| Separation Task | Stage | Text | | | | | | Visual | | |
|---|---|---|---|---|---|---|---|---|---|---|
| | | General SFX | Speech | Speaker | Music | Instr(wild) | Instr(pro) | General SFX | Speaker | Instr(wild) |
| SAJ (↑) | PT | 3.93 | 2.90 | 3.28 | 4.14 | 3.17 | 3.36 | - | - | - |
| | FT | **4.14** | **4.55** | **4.07** | **4.38** | **4.48** | **4.82** | - | - | - |
| CLAP (↑) | PT | **0.31** | 0.28 | **0.23** | **0.31** | 0.21 | 0.15 | - | - | - |
| | FT | 0.30 | **0.36** | 0.21 | **0.31** | **0.30** | **0.29** | - | - | - |
| IB (↑) | PT | - | - | - | - | - | - | **0.24** | 0.22 | 0.20 |
| | FT | - | - | - | - | - | - | **0.24** | **0.24** | **0.22** |
| AES-PC (↓) | PT | 2.57 | 2.91 | 2.98 | 4.53 | 3.54 | 4.72 | 3.09 | 3.49 | 3.69 |
| | FT | **2.08** | **1.89** | **1.94** | **4.44** | **3.16** | **3.24** | **2.22** | **2.20** | **3.26** |

*Table 20.* Effect of using pseudo-labeled audio stem data

| Separation Task | Setting | Text | | | | | | Visual | | |
|---|---|---|---|---|---|---|---|---|---|---|
| | | General SFX | Speech | Speaker | Music | Instr(wild) | Instr(pro) | General SFX | Speaker | Instr(wild) |
| SAJ (↑) | w/o PL | 4.02 | 4.50 | 4.06 | 4.34 | 4.34 | 4.80 | - | - | - |
| | PL | **4.14** | **4.55** | **4.07** | **4.38** | **4.48** | **4.82** | - | - | - |
| CLAP (↑) | w/o PL | **0.30** | 0.34 | **0.21** | **0.31** | 0.29 | 0.28 | - | - | - |
| | PL | **0.30** | **0.36** | **0.21** | **0.31** | **0.30** | **0.29** | - | - | - |
| IB (↑) | w/o PL | - | - | - | - | - | - | 0.23 | 0.21 | **0.22** |
| | PL | - | - | - | - | - | - | **0.24** | **0.24** | **0.22** |
| AES-PC (↓) | w/o PL | 2.36 | 1.95 | 2.02 | **4.38** | 3.19 | **3.24** | 2.28 | 2.29 | **3.25** |
| | PL | **2.08** | **1.89** | **1.94** | 4.44 | **3.16** | **3.24** | **2.22** | **2.20** | 3.26 |

evaluation baseline. We also provide several examples with different scores to enable few-shot learning.

These baselines cover the main paradigms of signal processing-based separation measures and general-purpose multimodal LLM-based metrics, allowing for a comprehensive comparison with our task-specific SAJ model. We report Pearson (PCC) and Spearman (SRCC) correlation between automatic metrics and human ratings.

As shown in Table 21, the proposed SAM Audio Judge model consistently outperforms all baselines across the three modalities, *i.e.* speech, music, and sound effects, under both PCC and SRCC. SAJ achieves markedly higher correlations with human ratings, reaching PCCs of 0.883, 0.815, and 0.815 for speech, music, and sound, respectively, and SR-CCs of 0.817, 0.714, and 0.781. In contrast, baseline models such as CLAP and Gemini-2.5-pro show moderate correlations, while distortion-based metric (SDR Estimator) fails to capture perceptual quality, often yielding low or even negative correlations. The consistent performance of SAJ across modalities highlights its ability to generalize beyond speech to more complex acoustic domains such as music and environmental sounds. These results confirm that the proposed SAJ model effectively captures human perceptual judgment by leveraging joint audio–text representations and text-conditioned pretraining, offering a more reliable and fine-grained evaluation framework than existing baselines.

### D.10.2. JUDGE AS A RERANKER

To evaluate the impact of different rerankers, we apply them to the SAM Audio model, which generates eight candidate separation outputs for each input mixture. Each reranker is responsible for selecting the best candidate according to its scoring mechanism, and the selected outputs are then evaluated on the test set (see Table 10). We compare three configurations: Judge, CLAP, and their combination (CLAP w/ Judge). The combined reranker computes a linear combination of the two scores with a weighting ratio of 5:1 (CLAP : Judge).

Table 22 presents a comparison of NWR between different reranking configurations across five separation tasks: speech, speaker, music, instrument, and general sound separation. Overall, we observe that integrating the SAM Audio Judge model as a reranker consistently improves or stabilizes performance relative to using CLAP alone.

When comparing Judge vs. CLAP, the Judge reranker achieves higher NWR in most categories (e.g., 0.17 vs. 0.19 in speech and 0.15 in sound separation), indicating its stronger ability to align selection decisions with perceptual quality. After combining both signals (CLAP w/ Judge vs. CLAP), the hybrid reranker further improves NWR on music (0.14) and instrument (0.03) separation, suggesting that CLAP score can be complementary to SAJ. Finally, when CLAP w/ Judge is evaluated against Judge directly,

*Table 21.* Comparison Between SAM Audio Judge Model and Baselines

| Model | Speech | | | | Music | | | | Sound | | | |
|---|---|---|---|---|---|---|---|---|---|---|---|---|
| | Overall | Recall | Precision | Faithfulness | Overall | Recall | Precision | Faithfulness | Overall | Recall | Precision | Faithfulness |
| Pearson Correlation Coefficient (PCC) | | | | | | | | | | | | |
| CLAP | 0.490 | 0.431 | 0.283 | 0.477 | 0.487 | 0.416 | 0.385 | 0.432 | 0.367 | 0.431 | 0.283 | 0.418 |
| SDR Estimator | 0.336 | 0.004 | 0.403 | 0.055 | 0.369 | 0.157 | 0.388 | 0.182 | 0.181 | 0.040 | 0.222 | 0.055 |
| Gemini-2.5-pro | 0.487 | 0.498 | 0.169 | 0.430 | 0.351 | 0.287 | 0.115 | 0.303 | 0.462 | 0.493 | 0.192 | 0.369 |
| **SAM Audio Judge** | **0.883** | **0.943** | **0.841** | **0.891** | **0.815** | **0.858** | **0.766** | **0.791** | **0.815** | **0.837** | **0.775** | **0.818** |
| Spearman Rank Correlation Coefficient (SRCC) | | | | | | | | | | | | |
| CLAP | 0.380 | 0.291 | 0.325 | 0.273 | 0.285 | 0.293 | 0.199 | 0.296 | 0.493 | 0.376 | 0.388 | 0.406 |
| SDR Estimator | 0.338 | 0.000 | 0.395 | 0.079 | 0.390 | 0.203 | 0.375 | 0.210 | 0.173 | 0.053 | 0.220 | 0.073 |
| Gemini-2.5-pro | 0.495 | 0.361 | -0.015 | 0.117 | 0.338 | 0.232 | 0.010 | 0.008 | 0.390 | 0.324 | -0.006 | 0.180 |
| **SAM Audio Judge** | **0.817** | **0.573** | **0.774** | **0.573** | **0.714** | **0.569** | **0.658** | **0.476** | **0.781** | **0.660** | **0.734** | **0.607** |

*Table 22.* Net Win Rate Comparison Between Rerankers (Reranker A vs. Reranker B)

| Reranker A | Reranker B | Speech | Speaker | Music | Instr(wild) | General |
|---|---|---|---|---|---|---|
| SAJ | CLAP | 0.17 | 0.19 | 0.09 | 0.01 | 0.15 |
| CLAP w/ SAJ | CLAP | 0.18 | 0.18 | 0.14 | 0.03 | 0.06 |
| CLAP w/ SAJ | SAJ | 0.00 | 0.02 | 0.10 | 0.15 | 0.07 |

the NWR drops generally compared to its evaluation against CLAP (e.g., 0.00 in speech, 0.02 in speaker, and 0.10–0.15 in music/instrument), showing that SAJ score already captures most of the reranking signal.

In summary, the results suggest that the Judge-based reranker is highly effective across audio domains. Combining judge and CLAP produces largest gains, particularly in speech and general sound separation, where reranking with CLAP alone tends to underperform.

*Table 23.* Comparison of representative audio separation benchmarks vs. SAM AUDIO-BENCH. Real: Y/N/M (mixed real+synthetic). Prompts: T=Text, V=Visual, S=Temporal spans.

| Benchmark | Real | Task coverage | | | | | Prompt(s) | Source / mixtures |
|---|---|---|---|---|---|---|---|---|
| | | Sp. clean | Spkr sep. | General | Mus. clean | Instr. stems | | |
| WSJ0-2mix / WHAM! / WHAMR![a] | N | – | ✓ | – | – | – | – | Synthetic 2-spk mixtures from WSJ0; WHAM/WHAMR add real noise |
| LibriMix[b] | N | – | ✓ | – | – | – | – | Synthetic speech mixtures from LibriSpeech + noise |
| DNS / VoiceBank+DEMAND[c] | M | ✓ | – | – | – | – | – | Noisy speech from VoiceBank, DNS challenge (synthetic + some real) |
| FUSS[d] | N | – | – | ✓ | – | – | – | Synthetic mixtures from diverse sound event stems |
| MUSDB18 / MDX[e] | M | – | – | – | – | ✓ | – | Studio multitrack music; vocals/drums/bass/other stems |
| Slakh2100[f] | N | – | – | – | – | ✓ | – | Fully synthetic MIDI-rendered multitrack music |
| MedleyDB / URMP[g] | M | – | – | – | – | ✓ | – | Real and studio multitrack ensembles, instrument stems |
| AudioSep (AudioSet / AudioCaps / Clotho mixtures)[h] | N | ✓ | – | ✓ | ✓ | ✓ | T | Synthetic mixtures from AudioSet, AudioCaps, Clotho, etc. |
| DCASE lang-queried source sep.[i] | M | – | – | ✓ | – | – | T | Synthetic mixtures from captioned / labeled sound events (e.g., FSD50K/AudioSet-style) |
| CLAP-based text-queried sep.[j] | N | – | – | ✓ | – | – | T | Synthetic mixtures from AudioSet / FSD50K-style datasets |
| AVSpeech-based AV speech sep.[k] | M | ✓ | ✓ | – | – | – | V | Synthetic 2-spk mixtures, sources from AVSpeech, LRS2/3, VoxCeleb |
| MUSIC[l] | M | – | – | – | – | ✓ | V | Real online performance videos; synthetic or curated multi-instrument mixtures |
| AVSBench / LU-AVS[m] | M | – | – | ✓ | – | ✓ | V,S | Real YouTube-like videos with object/instrument sounds, AV masks/spans |
| **SAM AUDIO-BENCH (ours)** | **Y** | ✓ | ✓ | ✓ | ✓ | ✓ | **T,V,S** | **Real in-the-wild A/V from AudioSet, VGGSound, MUSIC, AVSpeech, CondensedMovies** |

[a] WSJ0-2mix; WHAM!/WHAMR! (Hershey et al., 2016; Wichern et al., 2019; Maciejewski et al., 2020).

[b] LibriMix (Cosentino et al., 2020).

[c] VoiceBank+DEMAND; DNS challenge datasets (Valentini-Botinhao, 2017; Reddy et al., 2021).

[d] FUSS (Wisdom et al., 2021).

[e] MUSDB18/HQ; SiSEC/MDX (Rafii et al., 2017; Mitsufuji et al., 2022; Fabbro et al., 2024).

[f] Slakh2100 (Manilow et al., 2019).

[g] MedleyDB; URMP (Bittner et al., 2014; Li et al., 2018).

[h] AudioSep benchmark from mixtures of AudioSet, AudioCaps, Clotho, etc. (Liu et al., 2022; Kim et al., 2019; Drossos et al., 2020; Gemmeke et al., 2017).

[i] DCASE language-queried source separation benchmark (e.g., DCASE 2024 Task 8).

[j] CLAP-based text-queried separation benchmarks (Ma et al., 2024).

[k] AV speech separation on AVSpeech, LRS2/LRS3, VoxCeleb (Ephrat et al., 2018; Afouras et al., 2018; Chung et al., 2018).

[l] MUSIC is a canonical AV instrument separation dataset, used by Zhao et al. (2018).

[m] AVSBench; LU-AVS (Zhou et al., 2022; Liu et al., 2024a).

Abbrev.: Real Y/N/M; Sp. clean = speech cleaning; Spkr sep. = speaker separation; General = open-domain / general sound separation; Mus. clean = music cleaning; Instr. stems = instrument stem separation; Prompts: T=text, V=visual, S=temporal spans.

