# OpenReview forum: "SAM Audio: Segment Anything in Audio"
_ICML.cc/2026/Conference — ICML 2026 regular_

### Official Review · Reviewer_nHCd · 2026-03-06

**Soundness:** 4
**Presentation:** 3
**Significance:** 4
**Originality:** 3
**Overall Recommendation:** 5
**Confidence:** 5

**Summary:**

This paper proposes SAM-Audio, a foundation model for general audio source separation that supports multimodal prompting, including text descriptions, visual masks, and temporal spans. The model is built on a Diffusion Transformer (DiT) architecture, operates in a DAC-VAE latent space, and is trained using flow matching on a large-scale dataset compiled via synthetic mixing and a scalable pseudo-labeling pipeline. Additionally, the paper introduces a multimodal, in-the-wild separation benchmark called SAM-Audio-Bench, and a reference-free evaluation model named SAM-Audio-Judge (SAJ) that correlates strongly with human perceptual judgments. Experimental results demonstrate that SAM-Audio achieves state-of-the-art performance across multiple audio domains, including general sound effects, speech, and music, often outperforming both general-purpose and specialized models.

**Compliance With Llm Reviewing Policy:**

Affirmed.

**Final Justification:**

Response forwarded by Area Chair solved my concerns.

**Key Questions For Authors:**

Please ensure all formatting issues are resolved for the camera-ready version.

**Limitations:**

yes

**Strengths And Weaknesses:**

Strengths:
1. Integrating text, visual, and temporal span prompting into a single generative framework is a highly novel and practical approach to audio separation. The introduction of span prompting for frame-level temporal control, and using automated span prediction to boost text-prompted separation, are particularly contributive.
2. The release of SAM-Audio-Bench and the SAM-Audio-Judge (SAJ) evaluation metric provides valuable resources for the community, moving evaluation methodologies beyond synthetic mixtures and reference-based metrics (such as SI-SDR) that often correlate poorly with human perception.
3. The methodology is technically rigorous. The authors conduct extensive evaluations across diverse domains against strong baselines. The use of human evaluation via a side-by-side protocol strengthens the empirical claims.

Weaknesses:
1. The paper suffers from several formatting issues that must be fixed. Multiple URLs in the references overflow the column margins (e.g., lines 495-500 on page 9). On page 28, the footnotes for Table 22 exceed the bottom margin and overlap with the ICML confidentiality footer, making the text illegible.
2. The paper acknowledges that visual prompting yields lower subjective scores compared to text prompting. This suggests that visual grounding remains a challenging bottleneck, possibly due to the simple fusion mechanism (element-wise addition) or noisy training data.

---

### Official Review · Reviewer_sj7T · 2026-03-09

**Soundness:** 2
**Presentation:** 3
**Significance:** 3
**Originality:** 3
**Overall Recommendation:** 4
**Confidence:** 4

**Summary:**

This paper presents SAM AUDIO, a generative foundation model for general-purpose audio source separation built upon a Diffusion Transformer architecture and trained with flow matching. The model's primary contribution is unifying text descriptions, visual masks, and temporal span prompts into a single framework, allowing users to flexibly isolate specific target sounds and their corresponding residuals from complex mixtures. To achieve robust open-domain performance across speech, music, and general sound effects, the authors designed a scalable data engine that utilizes fully-real triplets, synthetic mixtures, and pseudo-labeled stems bootstrapped from in-the-wild recordings. Furthermore, to address the lack of realistic and comprehensive evaluation setups, the paper introduces SAM AUDIO-BENCH, a multimodal in-the-wild benchmark, along with SAM AUDIO-JUDGE, a reference-free metric fine-tuned to align with human perceptual judgments.

**Compliance With Llm Reviewing Policy:**

Affirmed.

**Key Questions For Authors:**

Please refer to the concerns and suggestions detailed in the Weaknesses section.

**Limitations:**

yes

**Strengths And Weaknesses:**

## Strengths

Significance & Originality: Unifying text, visual, and temporal span prompts within a single Flow-Matching Diffusion Transformer framework is a highly practical and creative combination of multimodal technologies. The introduction of span prompting, in particular, provides a novel and much-needed mechanism for fine-grained temporal control over audio extraction.

Soundness: The data curation pipeline is meticulously designed and highly robust. The authors successfully combine fully-real triplets, domain-aware synthetic mixtures, and a rigorously filtered pseudo-labeling engine to overcome the inherent scarcity of high-quality isolated audio stems.

Contribution: The creation and release of SAM AUDIO-BENCH is a significant contribution to the community, offering a realistic, in-the-wild evaluation dataset with comprehensive multimodal annotations.

Presentation: The paper is clearly written, and the overall narrative is fluent and easy to follow.

## Weaknesses

Soundness (Metric Leakage and Baseline Comparison): There is a potential fairness concern in the experimental setup that could affect the interpretation of the State-of-the-Art claims. During inference, SAM AUDIO generates 8 candidate outputs and reranks them to select the final result using a linear combination of the SAM Audio Judge (SAJ) and CLAP scores. The model is subsequently evaluated against baselines that presumably perform standard single-pass inference. Relying on the exact same SAJ and CLAP metrics for both the inference search heuristic and the final evaluation introduces a form of metric leakage. This approach essentially allows the model to optimize directly for the test metrics, giving it an inherent architectural advantage over comparative baselines that do not employ this reranking strategy.

---

### Official Review · Reviewer_8iVu · 2026-03-09

**Soundness:** 2
**Presentation:** 1
**Significance:** 2
**Originality:** 3
**Overall Recommendation:** 4
**Confidence:** 4

**Summary:**

The authors propose SAM Audio, a general-purpose audio separation model that can isolate a target sound from a mixture using a combination of text prompts, visual masks, and temporal span prompts. SAM Audio is a flow-matching diffusion transformer operating in a DAC-VAE latent space. It jointly predicts both the target stem and the residual audio in a single pass, demonstrating state-of-the-art performance on various separation tasks.

**Compliance With Llm Reviewing Policy:**

Affirmed.

**Final Justification:**

The authors provided a late rebuttal that mostly addresses my concerns.

**Key Questions For Authors:**

- How do the authors foresee SAM Audio Bench to be used? It does not contain ground truth stems, why did the authors not use a part of their clean-separated audio data as part of the benchmark? Having ground truth stems to compare against would help greatly.
- Where does the multi-track music and conversational speech training data come from?
- How are ambiguous prompts found during benchmark filtering?
- How are the visual masks generated? The manuscript mentions at various points either SAM, SAM2, SAM3 or an in-house text-prompted video segmentation model.

**Limitations:**

no dedicated limitation section is provided

**Strengths And Weaknesses:**

In general I appreciate the author's effort to push stem-agnostic source separation. The model performs well across all benchmarks and the provided SAM Audio Judge is also a welcomed addition. I'm fairly surprised that SAM Audio is able to outperform domain-specific models such as HTDemucs on music instrument separation.

The authors write
> Within each modality, we filter out samples with ambiguous prompts that could map to multiple plausible sources

I'm a bit concerned about this. Isn't this a fundamental limitation? The difference between this "cleaned" setting and messy in-the-wild data is not addressed as far as I can tell. The performance during evaluation might give the wrong impression of how well this approach actually works on in-the-wild data where ambiguity is often present.

The use of Big O notation to report dataset sizes in Tables 2 and 4 (e.g., O(100), O(10), O(1)) is unconventional and technically incorrect, since Big O absorbs constant factors, making all these entries equivalent to O(1). Simply reporting approximate counts or order-of-magnitude ranges (e.g., ~100M, ~10K) would be clearer and consistent with standard practice.

Also, the manuscript needs some love. To name a few things:
- the table captions are placed incorrectly most of the time according to ICML style guide.
- the URL of the clear voice reference is out of bounds
- inconsistent styling (Fig. X vs. Figure X)
- line 227 right column refers to SAM3 is not referenced
- on page 25 there is a missing reference "Appendix ??"
- page 28 is going out of bounds

---

### Official Review · Reviewer_WcP2 · 2026-03-12

**Soundness:** 3
**Presentation:** 3
**Significance:** 3
**Originality:** 3
**Overall Recommendation:** 5
**Confidence:** 4

**Summary:**

This paper highlights four main challenges in building a unified foundation model for general audio source separation, achieving cross-domain generalization across speech, music, and environmental sounds,  unifying multi-modal prompts,  enabling precise temporal control; and performing reliable evaluation without clean references. To address these challenges, the authors propose SAM Audio. The model adopts a flow-matching diffusion transformer architecture to better capture the one-to-many nature of audio separation. It also unifies text, visual, and temporal span prompts within a single framework, allowing flexible user interaction. In addition, the proposed span prompting mechanism introduces adjustable temporal intervals to help disambiguate overlapping sound events. Finally, the paper presents SAM Audio Judge, a reference-free evaluation model designed to align closely with human perceptual judgments. Experiments demonstrate that SAM Audio achieves state-of-the-art performance across text, visual, and span-prompted separation tasks, outperforming both general-purpose and specialized baselines on datasets from real-world and professional audio domains.

**Compliance With Llm Reviewing Policy:**

Affirmed.

**Final Justification:**

The paper is technically sound with a meaningful contribution and solid empirical evaluation. The authors did not provide a rebuttal, and my assessment remains unchanged. Therefore, I maintain my initial recommendation of acceptance.

**Key Questions For Authors:**

Q1: Is there an explicit sum-to-mixture constraint?
The joint generation of target and residual stems is described, but given that the original mixture x_mix is always available as input, I'm unclear whether the model enforces xtgt + xres = x_mix as a hard constraint, or if this relationship is learned implicitly from training data.
For phase-sensitive signals, even small deviations from this constraint could introduce audible artifacts. Does the flow-matching objective or the DiT architecture provide any implicit regularization to maintain mixture consistency?

Q2: How does visual prompting handle off-screen or occluded sounds?
When an object leaves the frame or becomes occluded, the visual input M may become uninformative (zero or missing). Since the audio mixture x_mix is still provided, the model presumably falls back to audio-only separation. But for reappearing objects, does the visual mask act as a "reset" that overrides audio continuity, or is there implicit tracking across the occlusion?

Q3: How sensitive is the model to visual mask quality?
The current implementation uses SAM 2 to obtain pixel-level segmentation masks. It would be interesting to understand how much this level of precision actually contributes to performance. Have the authors considered testing simpler visual cues, such as bounding boxes, coarse masks, or even degraded segmentation? Since the paper reports that visual prompting performs worse than text prompting, such an ablation could help clarify whether the limitation comes from noisy audio-visual alignment or from the granularity of the visual input.

**Limitations:**

yes

**Strengths And Weaknesses:**

Strengths

S1: The empirical results are strong. SAM Audio achieves state-of-the-art performance across multiple audio separation domains, including general sound effects, speech, music, and instruments, and across different prompting modalities (text, visual, and temporal span). The gains over both general-purpose and specialized baselines are consistent. In addition, span prompting provides clear complementary benefits when combined with text prompts. The proposed evaluation model, SAM Audio Judge, also shows noticeably higher correlation with human perception compared with existing objective metrics.

S2: The data engineering pipeline is well designed. The pseudo-labeling strategy, which relies on an intermediate model checkpoint combined with multi-stage filtering (CLAP alignment, aesthetic scoring, and VAD), provides a practical way to scale training data without requiring manually annotated stems. This is an important step toward addressing the data scarcity problem in large-scale audio separation.

S3: The unified prompting framework is practically useful. By supporting text, visual, and temporal span prompts either individually or jointly, the system provides flexible control for a range of real-world use cases where a single prompt modality may not be sufficient.

Weaknesses

W1: Limited analysis of feature interaction. The model concatenates target and residual features along the channel dimension and relies on the Diffusion Transformer to model their interactions. However, the paper provides limited analysis of how the model handles potential interference between these channels. In particular, it is unclear whether the self-attention layers effectively mitigate cross-talk between target and residual signals. For example, when the visual prompt is weak or ambiguous, does the model tend to leak residual noise into the target stem?

W2: Missing appendix reference. In Section B.2.3 the text states that “Our prompts could be found in Appendix ??” with placeholder question marks instead of an actual section number. This appears to be an unedited LaTeX placeholder and suggests the manuscript was not fully proofread before submission.

W3: Formatting issue in references. The ClearVoice citation (lines 459–463) contains a URL that exceeds the right margin of the page. The bibliography formatting for long URLs should be adjusted to avoid overflow.

---

### Decision · Program_Chairs · 2026-04-30

**Decision:**

Accept (regular)

**Comment:**

This paper presents SAM Audio, an diffusion transformer based audio separation model that can use text, vision, or temporal span prompts to do the separation, all in a single framework. The key technical innovations are: i) a unified foundation model that incorporates the various input modalities for sound separation and ii) a training pipeline that jointly predicts the target audio and the residual from any combination of the input modalities. Experiments demonstrate state-of-the-art results.

The paper received overall positive reviews with reviewers acknowledging the empirical results, the training pipeline, and the unified prompting setup. There were concerns raised on the limited analysis of features and missing technical details, potential unfairness in the evaluations. Rebuttal from the authors addressed these concerns adequately, and all the reviewers recommended acceptance.

AC had an independent review of the paper and agrees with the reviewers that the approach to unifying modalities for a general-purpose audio foundation model is a substantial contribution. AC recommends acceptance. The authors must to incorporate the rebuttal responses into the main paper.